# Graph-Based Algorithms for Diverse Similarity Search

**Piyush Anand** [* 1]  **Piotr Indyk** [* 2]  **Ravishankar Krishnaswamy** [* 3]  **Sepideh Mahabadi** [* 3]  **Vikas C. Raykar** [* 1]
**Kirankumar Shiragur** [* 3]  **Haike Xu** [* 2]

## Abstract

Nearest neighbor search is a fundamental data structure problem with many applications. Although the main objective of the data structure is to quickly report data points that are closest to a given query, it has long been noted (Carbonell & Goldstein, 1998) that without additional constraints the reported answers can be redundant and/or duplicative. This issue is typically addressed in two stages: in the first stage, the algorithm retrieves a (large) number $r$ of points closest to the query, while in the second stage, the $r$ points are post-processed and a small subset is selected to maximize the desired diversity objective. Although popular, this method suffers from a fundamental efficiency bottleneck, as the set of points retrieved in the first stage often needs to be much larger than the final output. In this paper we present provably efficient algorithms for approximate nearest neighbor search with diversity constraints that bypass this two stage process. Our algorithms are based on popular graph-based methods, which allows us to "piggy-back" on the existing efficient implementations. These are the first graph-based algorithms for nearest neighbor search with diversity constraints. For data sets with low intrinsic dimension, our data structures report a diverse set of $k$ points approximately closest to the query, in time that only depends on $k$ and $\log \Delta$, where $\Delta$ is the ratio of the diameter to the closest pair distance in the data set. This bound is qualitatively similar to the best known bounds for standard (non-diverse) graph-based algorithms (Indyk & Xu, 2023). Our experiments show that the search time of our algorithms is substantially lower than that using the standard two-stage approach.

## 1. Introduction

Nearest neighbor search is a classic data structure problem with many applications in machine learning, computer vision, recommendation systems and other areas (Shakhnarovich et al., 2006). It is defined as follows: given a set $P$ of $n$ points from some space $X$ equipped with a distance function $D(\cdot, \cdot)$, build a data structure that, given any query point $q \in X$, returns a point $p \in P$ that minimizes $D(q, p)$. In a more general version of the problem we are given a parameter $k$, and the goal is to report $k$ points in $P$ that are closest to $q$. In a typical scenario, the metric space $(X, D)$ is the $d$-dimensional space, and $D(p, q)$ is the Euclidean distance between points $p$ and $q$.

Since for high-dimensional point sets the known *exact* nearest neighbor search data structures are not efficient, several approximate versions of this problem have been formulated. A popular theoretical formulation relaxes the requirement that the query algorithm must return the exact closest point $p$, and instead allows it to output any point $p' \in P$ that is a $c$-approximate nearest neighbor of $q$ in $P$, i.e., $D(q, p') \le cD(q, p)$. In empirical studies, the quality of the set of points reported by an approximate data structure is measured by its recall, i.e., the average fraction of the true $k$ nearest neighbors returned by the data structure.

Although minimizing the distance of the reported points to the query is often the main objective, it has long been noted (Carbonell & Goldstein, 1998) that, without additional constraints, the reported answers are often redundant and/or duplicative. This is particularly important in applications such as recommendation systems or information retrieval[1], where many similar variants of the same product, product seller, or document exist. For example, an update to the search results listing algorithm implemented by Google in 2019 ensures that "no more than two pages from the same site" are listed (Liaison, 2019). Such constraints can be formulated by assuming that each point is assigned a "color" (e.g., site id or product seller) and requiring the data

---

[*]Equal contribution  [1]Microsoft  [2]MIT  [3]Microsoft Research.  Correspondence to: Sepideh Mahabadi <smahabadi@microsoft.com>, Haike Xu <haikexu@mit.edu>.

*Proceedings of the 42$^{nd}$ International Conference on Machine Learning*, Vancouver, Canada. PMLR 267, 2025. Copyright 2025 by the author(s).

---

[1]More recent applications include Retrieval Augmented Generation (RAG) systems. For example, (Chen & Choi, 2025) argues that retrieved items need to be diverse with respect to perspective to excel in RAG queries. Similarly (Rezaei & Dieng, 2025) argues for diverse retrieval using Vendi scores.

structure to output a set $S$ of $k$ points containing *at most $k'$ points of each color*, whose distances to $q$ are (approximately) optimal. A more general formulation allows an arbitrary *diversity metric* $\rho$ (typically different from $D$), and requires the data structure to report a set $S$ of $k$ points such that for any distinct $p, p' \in S$, $\rho(p, p') \geq C$, for some required diversity parameter $C > 0$.

The aforementioned paper of (Carbonell & Goldstein, 1998) stimulated the development of the rich area of *diversity-based reranking*, which became the dominant approach to this problem. The approach proceeds in two stages. In the first stage, the data structure retrieves $r$ points closest to the query, where $r$ can be much larger than the desired output $k$. In the second stage, the $r$ points are post-processed to maximize the diversity objective of the reported $k$ points.

Though popular, the reranking approach to diversifying nearest neighbor search suffers from a fundamental efficiency bottleneck, as the algorithm needs to retrieve a large enough set to ensure that it contains the $k$ diverse points. In many scenarios, the number $r$ of points that need to be retrieved can be much larger than $k$ (see e.g., Figure 1 and the discussion in the experimental section). In the worst case, it might be necessary to set $r = \Omega(n)$ to ensure that the optimal set is found. This leads to the following algorithmic question:

> *Is it possible to bypass the standard reranking pipeline by directly reporting the $k$ diverse points, in time that depends on $k$ and not $r$?*

In this paper, we aim to solve this problem both in theory and in practice. Because of these dual goals, we focus on designing efficient *graph-based* algorithms for diverse similarity search. In graph-based algorithms, the data structure consists of a graph between the points in $P$, and the query procedure performs greedy search over this graph to find points close to the query. Graph-based algorithms such as HNSW (Malkov & Yashunin, 2018), NGT (Iwasaki & Miyazaki, 2018), and DiskANN (Jayaram Subramanya et al., 2019) have become popular tools in practice, often topping Approximate Nearest Neighbor benchmarks (Aumueller et al., 2024). In addition, they are highly versatile, as they do not put any restrictions on the distance function $D$. Although most of the work in this area is purely empirical, a recent paper (Indyk & Xu, 2023) gives approximation and running time guarantees for some of those algorithms.

## 1.1. Our Results

We give a positive answer to the aforementioned question, by designing a variant of the DiskANN algorithm that reports approximate nearest neighbors of a given query satisfying diversity constraints. Our theoretical analysis assumes the same setup as in (Indyk & Xu, 2023). Specifically, we assume that the input point set $P$ has bounded *doubling dimension[2] $d$*, and that its aspect ratio (the ratio of the diameter to the closest pair distance) is at most $\Delta$. Under this assumption, we show that the query time of our data structures is polynomial in $k$, $\log n$ and $\log \Delta$.

Formally, our result is as follows. (Here we state the result in the simplest setting, where the diversity is induced by point colors and $k' = 1$. (See Theorem A.5 for the general result statement.)

**Theorem 1.1.** *Consider the data structure constructed by Algorithm 1. Given a query $q$, let $R$ be the radius of the smallest ball around $q$ (w.r.t. the metric $D$) which contains $k$ points of different colors. Then the search Algorithm 2 returns a set $S$ of $k$ points of different colors such that, for all $p \in S$,*

$$D(q, p) \leq \left( \frac{\alpha + 1}{\alpha - 1} + \epsilon \right) R.$$

*The search algorithms takes $O\left(k \log_\alpha \frac{\Delta}{\epsilon}\right)$ steps, where each step takes $\tilde{O}\left(k(8\alpha)^d \log \Delta\right)$ time. The data structure uses space $O(nk(8\alpha)^d \log \Delta)$.*

We note that the approximation factor with respect to $D$, as well as the running time bounds, are essentially the same as the bounds obtained in (Indyk & Xu, 2023) for the non-diverse approximate nearest neighbor problem. The main difference is that the bound in (Indyk & Xu, 2023) does not depend on $k$, as they consider only the case of $k = 1$.

As mentioned earlier, Theorem 1.1 generalizes to arbitrary $k'$ and general diversity metric $\rho$, as discussed later.

**Experimental results.** We adapt our theoretical algorithms to devise fast heuristics based on them, and show the efficiency of our algorithms on several realistic scenarios. In one of them, we consider the task of showing ads to a user based on their search queries. Given a number $k$, say 100, of available slots, the goal is to choose ads from a large corpus, such that the *sellers* (i.e., colors) of those ads are all different. A more relaxed constraint requires that the number of ads shown from a single seller be bounded by at most $k'$, say 10. To achieve 95% recall@100 on this real-world scenario, the prevailing baseline approach of retrieving a much larger number of candidates using a regular ANN index and then choosing the best diversity-preserving $k$-sized subset of them has latency upwards of 8ms; our algorithm, on the other hand achieves a similar recall at a latency of around 1.5ms, resulting in an improvement upwards of **5X**! A production-quality implementation of our algorithm is currently under development for serving large-scale workloads at one of the major technology companies.

The key idea in our work - modifying graph construction to encourage diversity - is conceptually simple and broadly

---

[2]Doubling dimension is a measure of the intrinsic dimensionality of the pointset - see Preliminaries for the formal definition.

applicable. In principle, it can be integrated into any graph-based ANN algorithm that uses greedy search, such as HNSW, as it does not rely on DiskANN-specific internals. We chose to apply our ideas within the DiskANN framework because it is, to our knowledge, the only real-world graph-based ANN algorithm that comes with worst-case theoretical guarantees. This allows us to ground our contributions in a rigorous yet practically relevant setting.

**Generalizations.** On the theoretical side, we extend our results in several directions listed below. These are shown and proved in Section A.

- **Relaxing the diversity requirement.** First, in some applications, the requirement that *all* reported results have different colors is too strong. (For example, the aforementioned application to search (Liaison, 2019) allows for two points having the same color.) Therefore, we consider a more general constraint, requiring that no color should appear more than $k'$ times. We show how our results can be generalized to any $1 \leq k' \leq k$.

- **Generic metric $\rho$.**[3] Second, we generalize our results to the case where diversity is defined according to a given metric $\rho$ (also defined over $X$ but potentially different from $D$). Here, given a required diversity parameter $C$, the goal is to report a set $S$ of $k$ closest points to the query such that $min_{p_1,p_2 \in S}\rho(p_1,p_2) \geq C$. We say that such a set $S$ is $C$-*diverse*. Note that the color version is the special case where $\rho(p_1, p_2)$ is defined to be 0 if $p_1$ and $p_2$ are of the same color, and 1 otherwise.

- **Unifying the two generalizations.** In order to unify the above two results, and incorporate the notion of $k'$ into the generic metric $\rho$, we allow for each point in the reported set $S$ to be "similar" to at most $k' > 1$ other points in $S$. More formally, for any $p \in S$ there should be at most $k' - 1$ other points $p' \in S$ such that $\rho(p,p') < C$. We say that such a set $S$ is $(k',C)$-*diverse*[4]. We show how our algorithms can be modified to this most general formulation of the problem.

- **Primal vs Dual formulations.** Finally, instead of asking for the closest $k$ points to the query satisfying a diversity requirement parameterized by $C$, which we refer to as the *primal* variant of the problem, one can ask the *dual* question: Given a radius $R$, find a set of $k$ points within radius $R$ of the query, while maximizing the diversity. We show algorithms for both the primal and dual variants of the most general formulation of the problem.

---

[3]In fact our algorithms work for $\rho$ being any *pseudo-metric* allowing $\rho(p_1, p_2) = 0$ for two different points $p_1, p_2 \in P$, but for simplicity we refer to it as metric, throughout the paper.

[4]We note that the notion of $(k', C)$-diverse set is a new notion of diversity that strictly generalizes the widely used minimum-pairwise-distance notion for diversity.

## 1.2. Related Work

**Diverse nearest neighbor:** Nearest Neighbor Search with diversity requirement has been previously studied in the work of (Abbar et al., 2013a;b), where they presented a "diversified version" of the *Locality-Sensitive Hashing (LSH)* approach due to (Indyk & Motwani, 1998). However, their diversification approach does not carry over to the graph-based methods. Moreover, they provide a bi-criteria approximation only for the *dual formulation* of the problem, and do not consider the primal formulation. Finally, the distance functions $D$ that they consider are limited to Hamming distance or its variants like the Jaccard similarity (Abbar et al., 2013a). Although it is plausible that the result could be extended to other distances that are supported by LSH functions, not all distance functions satisfy this constraint.

**Graph-Based algorithms for similarity search:** This work focuses on algorithms that employ graph-based data structures for (approximate) nearest neighbor search. These approaches are applicable to general metric spaces, where, during preprocessing, the metric is approximated by constructing a carefully designed graph. Given a query point, the algorithm performs a greedy traversal of the graph to locate nearby points. Graph-based methods have received significant attention in both theoretical (Krauthgamer & Lee, 2004; Beygelzimer et al., 2006) and practical contexts (Malkov & Yashunin, 2018; Fu et al., 2019; Jayaram Subramanya et al., 2019). Further improvements can be obtained by optimizing the graph traversal during greedy search by either ruling out expensive distance calculations of candidates which will not improve the priority queue (by cleverly using estimates (Chen et al., 2023), and probabilistically excluding bad candidates using probabilistic partitioning methods (Lu et al., 2024)). For broader surveys of this area, see (Clarkson et al., 2006; Wang et al., 2021).

A related but different line of research studies nearest neighbor search with local attribute constraints (Wang et al., 2023). In (Wang et al., 2023), the goal is to obtain results with similar metadata as the query metadata (say color=RED or brand=NIKE as specified by the user query in a shopping scenario). In contrast, in our setting, there is no user specified constraint and the goal is to output relevant vectors with sufficiently dissimilar metadata. This crucial difference extends to both the problem formulation as well as the indexing and search strategies.

## 2. Preliminaries

**Problem definition.** Let $(X, D)$ be the underlying metric space, with distance function $D$. Let $P$ be our colored point set. For $p \in P$, we use $col[p]$ to denote its *color*.

**Definition 2.1** (colorful)**.** A set $S$ is *colorful* if no two points in $S$ have the same color.

**Definition 2.2** ($k'$-colorful). A set $S$ is $k'$-*colorful*, if within the multi-set $\{col[p_1], ..., col[p_k]\}$, no color appears more than $k'$ times.

Note that for $k'$-colorful for $k' = 1$ is equivalent to the colorful notion.

**Definition 2.3.** Given a subset $S \subset P$ of size $k$, and a query $q$, for each $i \leq k$, we use $S_i(q)$ to denote the distance of the $i$th closest point in $S$ to $q$. When $q$ is clear from the context, we drop $q$ and simply use $S_i$.

**Definition 2.4** (Colorful NN). Given a colored point set $P$, the goal of *colorful NN* is to preprocess $P$ and create a data structure such that given a query point $q$, one can quickly return a colorful subset $S \subset P$ of size $k$ such that $S_k$ is minimized.

Note that, when $k = 1$, our Colorful NN degenerates to the standard nearest neighbor search problem.

**Definition 2.5** ($k'$-Colorful NN). Given a colored point set $P$, the goal of $k'$-*colorful NN* is to preprocess $P$ and create a data structure such that given a query point $q$, one can quickly return a $k'$-colorful subset $S \subset P$ of size $k$ such that $S_k$ is minimized.

**Balls, doubling dimension, and aspect ratio.** We use $B_D(p, r)$ to denote a ball centered at $p$ with radius $r$, i.e., $B_D(p, r) = \{u \in X : D(u, p) < r\}$. We will drop the subscript $D$ if the metric is clear from the context.

We say a point set $P$ has *doubling dimension* $d$ if for any point $p$ and radius $r$, the set $B(p, 2r) \cap P$ can be covered by at most $2^d$ balls of radius $r$.

**Lemma 2.6.** *Consider any point set $P \subset X$ with doubling dimension $d$.*

*For any ball $B(p, r)$ centered at some point $p \in P$ with radius $r$ and a constant $\alpha$, we can cover $B(p, r) \cap P$ using at most $m \leq O(\alpha^d)$ balls with radius smaller than $r/\alpha$, i.e. $B(p, r) \cap P \subset \bigcup_{i=1}^{m} B(p_i, r/\alpha)$ for some $p_1 \ldots p_m \in X$.*

We define $\Delta = \frac{D_{max}}{D_{min}}$ to be the *aspect ratio* of the point set $P$ where $D_{max}(D_{min}$, resp.) represents the maximal (minimal, resp.) distance between any two points in the point set $P$.

# 3. Algorithm for Colorful NN

In this section, for the sake of simplicity of presentation, we focus on the simplest setting where $k' = 1$, and $\rho$ is the binary metric. The binary setting corresponds to our main application of seller diversity, and the case of $k' = 1$ focuses on retrieving $k$ closest points from the data set such that *all* of them have different colors/sellers. The algorithm that handles the general setting is presented in Section A.

**Intuition behind the algorithm.** At a high level, the "slow pre-processing" algorithm of (Indyk & Xu, 2023) uses the following intuition when pruning the graph: If $u$ and $v$ are "much closer" to each other than to another point $p$, then it is not necessary to connect $p$ to both $u$ and $v$. This makes it possible to track the progress of the search procedure as it identifies points closer to the query point and use the doubling dimension to bound the degree of the graph and the total space. Our algorithm retains these insights, but also requires several new ones, as now we need to show that the search algorithms can progress *while maintaining a colorful solution*. On a high-level, this is established using the following two intuitions. First, if the colors of $u$ and $v$ are the same, then again there is no need to connect $p$ to both of them, and we can use the same pruning as before. Second, if $p$ is already connected to $k$ points $v_1, \cdots, v_k$, all of which are much closer to $u$ compared to $p$, and that they all have different colors, then again there is no need to connect $p$ to $u$. This is roughly because no solution would need more than $k$ points of different colors in a relatively small neighborhood. The main challenge in converting these intuitions into a formal argument is in showing that such a graph keeps enough edges for a greedy search algorithm to converge to an approximately optimal solution for the colorful NN problem.

## 3.1. The Preprocessing Algorithm

The indexing algorithm is shown in Algorithm 1. In order to decide on the vertices that $p$ should be connected to, we sort all vertices based on their distance from $p$ in the list $\mathcal{L}$ and start connecting $p$ to them. If we have already connected $p$ to a close-by vertex $u$ that has the same color as $v$, then we do not connect $p$ to $v$ anymore. The same holds if we have already connected $p$ to $k$ near-by vertices all with different colors. We keep track of this using the array $rep[u]$.

**Lemma 3.1.** *The graph constructed by Algorithm 1 has degree limit $O(k(8\alpha)^d \log \Delta)$.*

*Proof.* Let's first bound the number of points not removed by others, then according to Line 10 in Algorithm 1, the degree bound will be that times $k$.

We use $\mathsf{Ring}(p, r_1, r_2)$ to denote the points whose distance from $p$ is larger than $r_1$ but smaller than $r_2$. For each $i \in [\log_2 \Delta]$, we consider the $\mathsf{Ring}(p, D_{max}/2^i, D_{max}/2^{i-1})$ separately. According to Lemma 2.6, we can cover $\mathsf{Ring}(p, D_{max}/2^i, D_{max}/2^{i-1}) \cap P$ using at most $m \leq O((8\alpha)^d)$ small balls of radius $D(p, u)/(4\alpha) \geq \frac{D_{max}}{2^{i+2}\alpha}$. According to the pruning criteria in Line 9, within each small ball, there will be at most one such point $u$ remaining. This establishes the degree bound of $O(k(8\alpha)^d \log \Delta)$. $\qquad\square$

## 3.2. The Search Algorithm

Algorithm 2 shows the search algorithm for the colorful nearest neighbor search problem. The algorithm is a greedy

**Algorithm 1** Indexing algorithm for colorful NN

1: **Input**: A set of $n$ points $P = \{p_1, ..., p_n\}$; $k$ is the size of the output; $\alpha$ is the parameter used for pruning.
2: **Output**: A directed graph $G = (V, E)$ where $V = \{1, ..., n\}$ are associated with the point set $P$.
3: **for** each point $p \in P$ **do**
4:      Sort all points $u \in P$ based on their distance from $p$ and put them in a list $\mathcal{L}$ in that order.
5:      **while** $\mathcal{L}$ is not empty **do**
6:          $u \leftarrow \underset{u \in \mathcal{L}}{\operatorname{argmin}} D(u, p)$
7:          Initialize $rep[u] \leftarrow \{u\}$
8:          **for** each point $v \in \mathcal{L}$ in order **do**
9:             **if** $D(u, v) \leq D(p, u)/(2\alpha)$ **then**
10:               **if** $col[v]$ not shown in $rep[u]$ and $|rep[u]| < k$ **then**
11:                  $rep[u] \leftarrow rep[u] \cup v$
12:               **end if**
13:             remove $v$ from $\mathcal{L}$
14:             **end if**
15:          **end for**
16:          add edges from $p$ to $rep[u]$
17:          Remove $u$ from $\mathcal{L}$
18:      **end while**
19: **end for**

---

**Algorithm 2** Search algorithm for colorful NN

1: **Input**: A graph $G = (V, E)$ with $N_{out}(p)$ being the out edges of $p$; query $q$; number of optimization steps $T$.
2: **Output**: A set of $k$ points ALG.
3: Initialize ALG $= \{p_1, ..., p_k\}$ to be any set of $k$ points with different colors.
4: **for** $i = 1$ to $T$ **do**
5:      $p_k \leftarrow$ the furthest point in ALG from $q$.
6:      $U \leftarrow N_{out}(p_k)$ and sort $U$ based on their distance from $q$
7:      **for** each point $u \in U$ **do**
8:          **if** ALG $\setminus p_k \bigcup u$ is colorful **then**
9:             ALG $\leftarrow$ ALG $\setminus p_k \bigcup u$
10:             Break
11:          **end if**
12:      **end for**
13: **end for**
14: **Return** ALG

---

algorithm where at each step, the neighbors of the furthest point from the query are additionally considered as candidates and a greedy solution is constructed among the points in the current solution and these neighbors. Next, we analyze the search algorithm and finally prove Theorem 1.1.

**Proposition 3.2.** *Let* OPT $= \{p_1^*, ..., p_k^*\}$ *be a colorful solution with minimized* OPT$_k$ *(see Definition 2.3), and* ALG $= \{p_1, ..., p_k\}$ *be the current solution (both ordered by distance from $q$). If $p_k \notin$ OPT, there exists a point $p^* \in$ OPT $\setminus$ ALG such that ALG $\setminus p_k \bigcup p^*$ is colorful.*

*Proof.* Observe that throughout the search algorithm, we maintain the property that ALG is colorful. Note that ALG $\setminus p_k$ has $k-1$ different colors, and OPT has $k$ different colors. Thus there should be a point $p^* \in OPT$ whose color is different from all points in ALG $\setminus p_k$. Note that such $p^*$ cannot belong to ALG and thus belongs to OPT $\setminus$ ALG. $\square$

**Lemma 3.3.** *There always exists a point $p' \in N_{out}(p_k)$ (for $p_k$ as defined in Line 5) such that (i) ALG $\setminus p_k \bigcup p'$ is colorful; and (ii) $D(p', q) \leq D(p_k, q)/\alpha + $OPT$_k(1 + 1/\alpha)$*

*Proof.* According to Proposition 3.2, for any current solution ALG with $p_k \notin$ OPT, there exists a point $p^* \in$ OPT $\setminus$ ALG such that ALG $\setminus p_k \cup p^*$ is colorful. If there exists an edge from $p_k$ to $p^*$, replacing $p_k$ with $p^*$ is a potential update. We set $p' = p^*$ and $D(p', q) \leq$ OPT$_k$ satisfies

the distance upper bound above.

Otherwise, we let $u$ be the point where $p^*$ was removed when processing $u$ on line 9 in Algorithm 1. Because $p^*$ was not connected from $p_k$, either there exists a point in $rep[u]$ with the same color, or $rep[u]$ has already got $k$ points with different colors. In the first case, we can set $p'$ to be the point in $rep[u]$ with the same color. In the latter case, by pigeon hole principle, there always exists a color in $rep[u]$ not shown in ALG $\setminus p_k$. Therefore, we can find a desired $p' \in rep[u]$ and it is connected to $p_k$.

We have proved that the $p'$ we found satisfies the colorful criteria. Now we will bound its distance upper bound.

$$D(p', q) \leq D(p^*, q) + D(p', p^*)$$
$$\leq D(p^*, q) + D(p', u) + D(p^*, u)$$
$$\leq D(p^*, q) + D(p_k, u)/(2\alpha) + D(p_k, u)/(2\alpha)$$
$$\text{(Line 9 in Algorithm 1)}$$
$$\leq D(p^*, q) + D(p_k, u)/\alpha \leq D(p^*, q) + D(p_k, p^*)/\alpha$$
$$\text{(Because } u \text{ is ordered earlier than } p^* \text{ in Algorithm 1)}$$
$$\leq D(p^*, q) + D(p_k, q)/\alpha + D(p^*, q)/\alpha$$
$$\leq D(p_k, q)/\alpha + \text{OPT}_k(1 + 1/\alpha)$$

$\square$

*Proof of Theorem 1.1.* Regarding the running time, the total number of edges connected from any point in ALG is bounded by $|U| \leq O(k(8\alpha)^d \log \Delta)$. In each step, the algorithm first sorts all these edges connected from $p_k \in$ ALG and then checks whether each of them can be added to the new ALG set. The total time spent per step is bounded by $O(|U| \log |U|)$. The overall time complexity is $\tilde{O}\left(k(8\alpha)^d \log \Delta\right)$ per step.

To analyze the approximation ratio, at time step $t$, we use $\mathsf{ALG}^t = \{p_1^t, ..., p_k^t\}$ to denote the current unordered solution. We denote $\mathsf{ALG}_k^t = \max_{i \in [k]} D(p_i^t, q)$. According to Algorithm 2 and Lemma 3.3, if $p_i$ is updated at time step $t$, we have $D(p_i^t, q) \leq D(p_i^{t-1}, q)/\alpha + \mathsf{OPT}_k(1 + 1/\alpha)$. By an induction argument, if a point $p_i$ is updated by $t$ times at the end of time step $T$, we have $D(p_i^T, q) \leq \frac{D(p_i^0, q)}{\alpha^t} + \frac{\alpha+1}{\alpha-1}\mathsf{OPT}_k$.

We now prove that $\mathsf{ALG}_k^T \leq \max_i \frac{D(p_i^0, q)}{\alpha^{T/k}} + \frac{\alpha+1}{\alpha-1}\mathsf{OPT}_k$. Let $i \in [k]$ be the index achieving the maximal distance upper bound. For the sake of contradiction, if $\mathsf{ALG}_k^T > \frac{D(p_i^0, q)}{\alpha^{T/k}} + \frac{\alpha+1}{\alpha-1}\mathsf{OPT}_k$, this means that $p_i^T$ was updated for at most $T/k - 1$ times. By a counting argument, there exists another index $j$ which was updated for at least $T/k + 1$ times. However, at the time $t$ when $p_j^t$ was already updated for $T/k$ times, $D(p_j^t, q) \leq \frac{D(p_j^0, q)}{\alpha^{T/k}} + \frac{\alpha+1}{\alpha-1}\mathsf{OPT}_k < \mathsf{ALG}_k^T \leq \mathsf{ALG}_k^t$, so the algorithm wouldn't have chosen $p_j^t$ to optimize because it couldn't have had the maximal distance at that time, leading to a contradiction. Therefore, we prove that $\mathsf{ALG}_k^T \leq \max_i \frac{D(p_i^0, q)}{\alpha^{T/k}} + \frac{\alpha+1}{\alpha-1}\mathsf{OPT}_k$.

Now we consider the following three cases depending on the value of the maximal $D(p_i^0, q)$. This case analysis is similar to the proof in Theorem 3.4 from (Indyk & Xu, 2023).

Case 1: $D(p_i^0, q) > 2D_{max}$.

Let $p_k^*$ be the point having the maximal distance from $q$ in an optimal solution $\mathsf{OPT}$. We know that for any $p_i^0$, we have $D(p_k^*, q) \geq D(p_i^0, q) - D(p_i^0, p_k^*) \geq D(p_i^0, q) - D_{max} \geq D(p_i^0, q)/2$. Therefore, the approximation ratio after $T$ optimization steps is upper bounded by $\frac{\mathsf{ALG}_k^T}{D(p_k^*, q)} \leq \frac{D(p_i^0, q)}{D(p_k^*, q)\alpha^{T/k}} + \frac{\alpha+1}{\alpha-1} \leq \frac{2}{\alpha^{T/k}} + \frac{\alpha+1}{\alpha-1}$. A simple calculation shows that we can get a $(\frac{\alpha+1}{\alpha-1} + \epsilon)$ approximate solution in $O(k \log_\alpha \frac{2}{\epsilon})$ steps.

Case 2: $D(p_i^0, q) \leq 2D_{max}$ and $\mathsf{OPT}_k > \frac{\alpha-1}{4(\alpha+1)}D_{min}$.

To satisfy $\frac{D(p_i^0, q)}{\alpha^{T/k}} + \frac{\alpha+1}{\alpha-1}\mathsf{OPT}_k \leq (\frac{\alpha+1}{\alpha-1} + \epsilon)\mathsf{OPT}_k$, we need $\frac{D(p_i^0, q)}{\alpha^{T/k}} \leq \epsilon\mathsf{OPT}_k$. Applying the lower bound $\mathsf{OPT}_k \geq \frac{\alpha-1}{4(\alpha+1)}D_{min}$, we can get that $T \geq k \log_\alpha \frac{2(\alpha+1)\Delta}{(\alpha-1)\epsilon}$ suffices.

Case 3: $D(p_i^0, q) \leq 2D_{max}$ and $\mathsf{OPT}_k \leq \frac{\alpha-1}{4(\alpha+1)}D_{min}$.

In this case, we must have $k = 1$, because otherwise $D(p_k^*, p_1^*) \leq 2D(p_k^*, q) < D_{min}$, violating the definition of $D_{min}$. Suppose $k = 1$ and the problem degenerates to the standard nearest neighbor search problem. After $T$ optimization steps, if $p_1^T$ is still not the exact nearest neighbor, we have $D(p_1^T, q) \geq D(p_1^T, p_1^*) - \mathsf{OPT}_1 \geq \frac{D_{min}}{2}$. Applying the upper bound of $D(p_1^T, q)$ and $\mathsf{OPT}_1$, we have $\frac{D_{min}}{2} \leq$

$D(p_1^T, q) \leq \frac{D(p_1^0, q)}{\alpha^T} + \frac{\alpha+1}{\alpha-1}\mathsf{OPT}_1 \leq \frac{D(p_1^0, q)}{\alpha^T} + \frac{D_{min}}{4}$. This can happen only if $T \leq \log_\alpha \frac{\Delta}{8}$.

In conclusion, $O(k \log_\alpha \frac{\Delta}{\epsilon})$ steps suffice to achieve the desired approximation ratio in Theorem 1.1. $\square$

We remark that it is easy to verify that for $k_1 \geq k_2$, the data structure built for the value $k = k_1$ can be used to search with respect to value $k = k_2$. Thus in applications, it suffices to only have an upper bound on the value of $k$ at the data structure construction time.

### 3.3. High-level Intuition about the Generalizations

Given our results on colorful NN, it is relatively simple to extend them to the $k'$-colorful NN version with the same bounds. *One key contribution is to demonstrate that, in the graph degree bound, the overhead factor $k$ can be reduced to $k/k'$ while preserving the approximation quality*. This reduces both the query time bound and the overall space used by the algorithm. This improvement is tight, in the following sense: When $k' = 1$, we recover the bound for colorful NN problem, and when $k' = k$, we recover the standard $k$-NN bound, where no additional factor is needed.

For our algorithm to work with a generic diversity metric $\rho$, we use an intuition similar to that in colorful case. However, instead of pruning an edge from the point $p$ to the point $u$ when $p$ is already connected to representative vectors $v_1, \ldots, v_k$ of different colors, we now choose the representatives based on the diversity metric $\rho$. We find a diverse subset of points in the neighborhood of $u$ (e.g., using the greedy Gonzales algorithm for the k-center problem) and connect $p$ only to those selected points $v_1, \ldots, v_k$. Again, the main challenge is to demonstrate that a greedy search algorithm can converge to an approximately optimal solution, given the set of edges we retain. The difficulty lies in the fact that we can only maintain an *approximately* diverse subset $\mathsf{ALG}$, in contrast to the colorful version, where we only needed $\mathsf{ALG}$ to contain $k$ different colors. As the algorithm proceeds with further iterations, the technical difficulty lies in ensuring that the approximation factor does not grow depending on the number of iterations.

## 4. Experimental Evaluation

In this section we provide an empirical evaluation of our methods[5]. To this end, we first devise a heuristic adaptation based on our provable algorithms for the $k'$-colorful NN problem as in Definition 2.5. As we note below, this problem itself captures several real-world notions of diversity.

At a high level, the difference between our heuristics

---

[5]The code is available at https://github.com/microsoft/DiskANN/tree/diversity

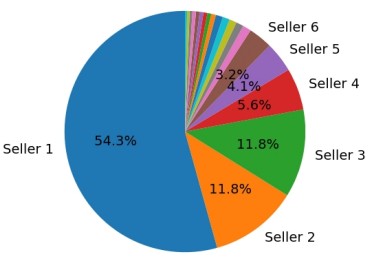

Figure 1: Seller distribution in a real-world dataset with 20 million base vectors, where the top 7 sellers constitute more than 90% of the data.

and our theoretical algorithms is similar to the difference between the fast- and slow-preprocessing algorithms in DiskANN (Jayaram Subramanya et al., 2019; Indyk & Xu, 2023)). Indeed, we deploy the same construction as in the fast-preprocessing variant of DiskANN, but modify the pruning procedure to insist that any node $u$ has sufficiently many colorful out-neighbors before an edge $(u, v)$ can get pruned, in addition to the geometric condition for pruning as in the original algorithm (Jayaram Subramanya et al., 2023). The number of colorful edges that are needed before pruning can occur is given by a tunable parameter $m$ in our algorithm, and indeed this is the direct heuristic analog of step 10 in Algorithm 1. This is a very high-level description, and we refer the interested reader to Appendix B for the complete pseudo-code of our heuristic algorithms.

Second, we run experiments using our heuristics on several different datasets, both real-world as well as synthetically generated. We show how our heuristic consistently delivers superior recall for a fixed latency budget, across datasets when compared to a natural baseline of using a vanilla DiskANN algorithm and enforcing diversity only via a final post-processing. We stress that both our real-world datasets are motivated from important shopping scenarios: the data points represent products and a color of a vector corresponds to either the seller or the brand of the product. It is then desirable to output results from a diverse set of sellers/brands (Liaison, 2019). Intuitively, displaying diverse results would lead to increased competition between the sellers, and also simultaneously higher click probabilities, thereby leading to an increase in revenue of the exchange.

### 4.1. Experiment Setup

All experiments were run on a Linux Machine with AMD Ryzen Threadripper 3960X 24-Core Processor CPU's @ 2.3GHz with 48 vCPUs and 250 GB RAM. All query throughput and latency measurements are reported for runs with 48 threads.

**Datasets.** We consider two real-world and three semi-synthetic datasets for evaluation.

*Real-world Seller Dataset:* Our first real-world seller dataset

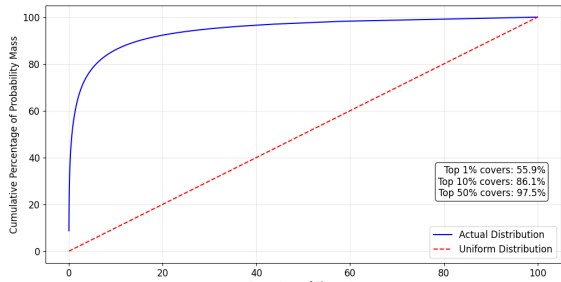

Figure 2: Brand cumulative distribution for Amazon dataset, showing the coverage of the vectors by the brands in sorted order. The top 10% of brands cover 86% of the vectors.

comprises of 64-dimensional vector embeddings of different products from a large advertisement corpus. Each product/vector is additionally associated with a *seller*, which becomes its color in our setting. There are 20 million base vectors, around 2500 sellers, and 5000 query vectors. This distribution is highly skewed, with an extremely small number (around 7) of sellers constituting more than 90% of the data, hence naturally motivating the need for enforcing diversity in the search results. The fraction of products corresponding to the top 20 sellers is shown in Figure 1.

*Amazon Automotive Dataset:* Our second real-world dataset is derived from the recently released Amazon dataset (Simhadri et al., 2024). It comprises of 384-dimensional vector embeddings of product descriptions listed on Amazon under the Automotive category. Each product/vector is additionally associated with a *brand*, which becomes the color. There are around 2 million base vectors and around 85000 brands. The distribution, while skewed, is far more balanced than the above seller dataset, with around 10% of the brands accounting for 80% of the vectors as summarized in Figure 2.

*"Skewed" Semi-synthetic Datasets:* We also consider the publicly available real-world Arxiv dataset (Embeddings, 2024) which contains OpenAI embeddings of around 2 million paper abstracts into 1536 dimensional vectors and the classical SIFT dataset of 1M vectors in 128 dimensions. These datasets do not contain any color information, so we synthetically add this information into the data set. Specifically, for the Arxiv dataset, we generate the color information as follows: for each vector, with probability 0.9, we assign a color selected from the set $\{1, 2, 3\}$ uniformly at random, and with 0.1 probability we assign a color selected uniformly at random from the set $\{4, \ldots, 1000\}$. Therefore the number of distinct colors is at most 1000 in this data set. For the SIFT dataset, we sampled one dominant colors with probability 0.8 and had a uniform distribution over 999 other colors with probability 0.2.

*"Balanced" Semi-synthetic Dataset:* Finally, we also consider another distribution which is globally uniform, but locally

skewed. Indeed, we use the same SIFT dataset for the vector data. For colors, we randomly partition the space into one thousand buckets, using a random hyperplane scheme. We then assign a unique *primary color* for each bucket. Now, each vector within any specific bucket is assigned its primary color with a high probability of $0.8$, and a uniformly random non-primary color with the remaining probability. It is then easy to see that the distribution is roughly balanced across colors from a global perspective, but quite skewed in any small local neighborhood.

**Tasks.** For all of the above datasets, we seek to retrieve $k = 100$ nearest neighbors. In one extreme scenario, we set $k' = 1$, i.e., all hundred of the returned results need to be of distinct colors. This type of setting would be relevant in a retrieval augmented generation setting where documents are typically chunked into several parts and each part is vectorized; when the user utters a query, we might want to retrieve the most relevant set of documents (as opposed to the most relevant chunks, which might all be from the same document) to a given query, before passing on these contents to a large-language model which then answers the user query. A natural way to enforce this document-level diversity during retrieval is to treat the document-id of any vector as its color, and using our diverse search routine.

In another scenario which might have more appeal in shopping or advertisement display, we seek to retrieve $k = 100$ nearest neighbors, while having $k' = 10$, i.e., no more than ten of the $k$ results may belong to any single color. This will promote the retrieval to display a diverse set of sellers/ brands in such scenarios, thereby hopefully increasing user satisfaction and engagement. This can also capture *intent diversity* in regular web-search (when we can use a simple classifier to represent the intent behind the data point as additional meta-data, which becomes the color in our setting – e.g., car or animal for different images of jaguar, ML or EE concept for transformer, etc.).

**Algorithms.** Since our algorithms are enhancements of the DiskANN algorithm, we use that as a natural baseline to compare against.

*Standard DiskANN Build + Post-Processing (Baseline):* In this baseline, we build a regular DiskANN graph without any diversity constraints. To answer a query, we first invoke the regular DiskANN search algorithm to retrieve $r \gg k$ candidates, again without any diversity constraints. Then we iterate over the retrieved elements in sorted order of distances to the query, and greedily include the ones which do not violate the $k'$ diversity constraint, until we have $k$ total elements. The parameter $r$ is tunable at search time, and higher $r$ yields more candidates, meaning more diverse candidates can be obtained using the post-processing step. However, higher $r$ also consumes more search complexity.

*Standard DiskANN Build + Diverse Search:* In this improvement, we use our diversity-preserving search Algorithm 7 discussed in the Appendix B, but the index construction remains the standard DiskANN algorithm.

*Diverse DiskANN Build + Diverse Search:* For our complete algorithm, we additionally use our diversity-aware index construction Algorithm 9 (Appendix B) which ensures sufficient edges are present to nodes of different colors in any neighborhood.

**Parameter setup.** For all of the above algorithms, we use fairly standard parameters of list-size $L = 200$ and graph-degree $64$ when building the graphs. During search, we vary the list-size $L$ at search time to get varying quality search results and plot the recall@100[6] vs average query latency.

### 4.2. Results

Our results are shown in plots of Figures 3, 4, and additionally in Figures 5, and 6 in Appendix C. As one can see, both of our algorithmic innovations play a crucial role in the overall search quality on the real-world dataset. For example, to achieve $95\%$ recall@100 in the real-world seller dataset, the baseline approach has latencies upwards of 8ms, while the improved search algorithm brings it down to $\approx 4.5$ms. Making both build and search diverse further brings it down to around $\approx 1.5$ms, resulting in an improvement of 5X.

The plot in Figure 4 reveals an interesting phenomenon: for high recalls (say $90\%$) on the semi-synthetic arXiv dataset, the post-processing approach has a latency of around 90ms, while the diverse search algorithm when run on the standard graph has a latency of around 135ms. This is perhaps because the standard graph construction might not have sufficiently many edges between nodes of different colors to ensure that the diverse search algorithm converges to a good local optimum. On the other hand, running the diverse search on the graph constructed keeping diversity in mind during index construction fares the best, with a latency of only around 25ms. A similar phenomenon occurs in the SIFT semi-synthetic dataset (shown in Appendix C) as well.

### 4.3. Build Diversity Parameter Ablation

In our heuristic graph construction algorithm (see Algorithms 8 and 9), the graph edges are added by considering *both the geometry of the vectors and the corresponding colors*. Loosely, the $\alpha$-pruning rule of DiskANN dictates

---

[6]Recall@100 is the size of the intersection of the algorithm's 100 returned results with the true 100 closest diverse candidates, averaged over all queries. The ground-truth set of top 100 diverse NNs for any query can be computed by iterating over all the vectors in sorted order of distances to the query, and greedily including the ones which do not violate the $k'$ diversity constraint, until we have accumulated $k$ total elements.

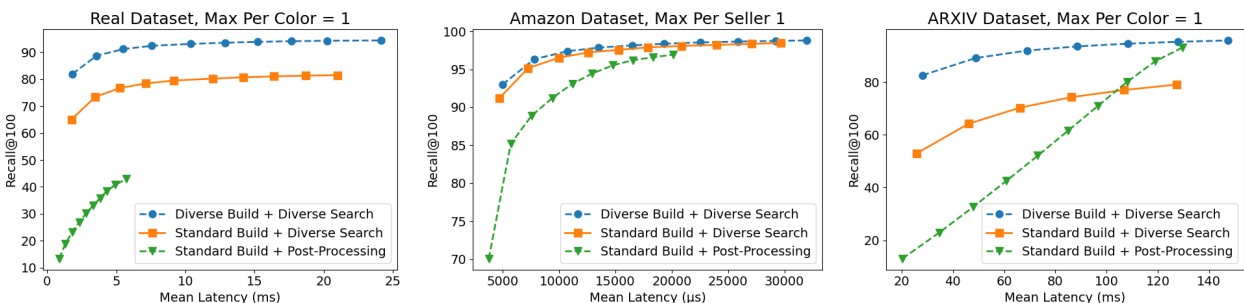

Figure 3: Recall vs Latency for real-world, Amazon, and ArXiv datasets with $k' = 1$.

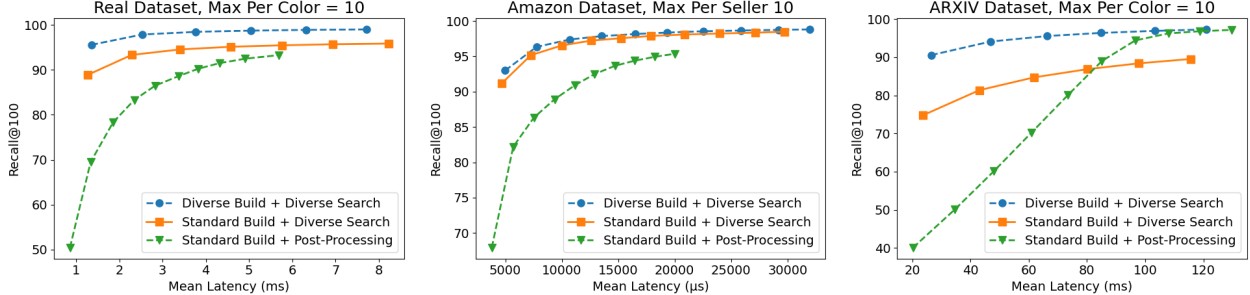

Figure 4: Recall vs Latency for real-world, Amazon, and ArXiv datasets with $k' = 10$.

| $m$ Parameter | Build Time (s) |
|:---:|:---:|
| 1 | 50 |
| 2 | 53 |
| 10 | 55 |

Table 1: Build Times w.r.t $m$ Parameter

that an edge $(u, v)$ is blocked by an existing edge $(u, w)$ if $d(w, v) \leq d(u, v)/\alpha$. In the original DiskANN algorithm, any edge $(u, v)$ which is blocked is not added. In our setting, we additionally enforce that *an edge needs to be blocked by edges of $m$ different colors* to not be added to the graph, where $m$ is a tuneable parameter. We now perform an ablation capturing the role of $m$ in the graph quality using the skewed SIFT dataset. Table 1 shows a table with build times for various indices by varying only the $m$ parameter, and Figure 7 in Appendix C shows the search quality of these different indices.

## 5. Conclusion

In this work, we introduced methods to incorporate diversity into the DiskANN graph-based similarity search algorithm. We proposed algorithms that address both binary diversity measures (the colorful case), as well as the more general case where an arbitrary metric is used to measure the diversity. Our algorithm can handle both the primal version—where the goal is to find nearest neighbors that meet a diversity requirement—and the dual version—where, given

a distance threshold, the goal is to retrieve a maximally diverse set of results within that threshold. Our algorithm can further allow a more relaxed diversity requirement parameterized using $k'$ allowing a limited number of similar items in the output. All our algorithms come with provable guarantees, and we further present heuristic adaptations that integrate easily with existing DiskANN implementations. We expect that similar strategies can be applied to other graph-based search methods, such as HNSW, to enhance result diversity. Finally, we demonstrated the practical effectiveness of our approach through experiments focusing on the binary diversity case across several application domains.

## Acknowledgements

Piotr Indyk was supported in part by the NSF TRIPODS program (award DMS-2022448). Haike Xu was supported by the Mathworks Fellowship. This work was conducted in part while Piotr Indyk and Sepideh Mahabadi were visitors at the Simons Institute for the Theory of Computing as part of the Sublinear Algorithms program.

## Impact Statement

This paper presents work whose goal is to advance the field of Machine Learning. There are many potential societal consequences of our work, none which we feel must be specifically highlighted here.

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

# A. Algorithms for the General Case

In this section we describe our algorithms for the most general case where $k'$ can take any value between $1$ and $k$, and the diversity metric $\rho$ is an arbitrary diversity metric. First we start with definitions, additional preliminaries, and our main theorem statements A.5 and A.6.

## A.1. Additional Preliminaries, Problem Formulation, and the Main Theorems

As before, we let $(X, D)$ be the underlying metric space, where $D$ measures the distance between the points. In this section, we assume that we are given a second metric space $(X, \rho)$ which measures the *diversity* between the points. As before $P$ is a subset of $n$ points in $X$. So for two points $p_1, p_2 \in P$, $\rho(p_1, p_2)$ measures their pairwise diversity.

Again, we use $B_D(p, r)$ (or just $B(p, r)$ for simplicity of exposition) to denote a ball centered at $p$ with radius $r$, i.e., $B_D(p, r) = \{u \in X : D(u, p) < r\}$. Similarly, we define the ball $B_\rho(p, r) = \{u \in X : \rho(u, p) < r\}$.

The following definitions recap the discussion in the introduction.

**Definition A.1** (*C*-diverse). Let $S$ be a set of points in $X$. We say $S$ is *C-diverse* if for any two points $p_1, p_2 \in S$, we have $\rho(p_1, p_2) \geq C$.

Note that the colorful setting, corresponds to the diversity metric $\rho$ being uniform. That is, we can set $\rho(p_i, p_j) = 0$ when $col[p_i] = col[p_j]$, and set $\rho(p_i, p_j) = 1$ otherwise. Then, we retrieve the colorful notion of diversity: a set $S$ of size $k$ is colorful iff it is 1-diverse. We further generalize this notion to allow the set to contain at most $k' > 1$ points that are similar to each other.

**Definition A.2** (($k', C$)-diverse). Let $S$ be a set of points in $X$. We say $S$ is ($k', C$)-diverse if for any point $p \in S$, we have $|B_\rho(p, C) \cap S| \leq k'$. Note that being $(1, C)$-diverse is equivalent to the notion of $C$-diverse.

We consider two dual variants of the diverse nearest neighbor search problem, both of which use two approximation factors: $c > 1$ is the "dissimilarity" approximation factor with respect to $D$, and $a > 1$ is the "diversity" approximation factor with respect to $\rho$.

**Definition A.3** (Primal Diverse NN). Given a point set $P$, diversity value $C$, and the value $k' \leq k$, the goal of *Primal Diverse NN* is to preprocess $P$ and create a data structure such that given a query point $q$, one can quickly return the *closest* set $S \subset P$ of size $k$ that is ($k', C$)-diverse. Here closeness of a set $S$ is measured by $S_k$ (See Definition 2.3).

In the approximate variant, for any $q \in X$, if OPT is a ($k', C$)-diverse set of $k$ points which minimizes $\mathsf{OPT_k}$, then the data structure outputs ALG that is ($k', C/a$)-diverse such that $\mathsf{ALG_k} \leq c \cdot \mathsf{OPT_k}$.

**Definition A.4** (Dual Diverse NN). Given a point set $P$, a radius $R$, and the value $k' \leq k$, the goal of *Dual Diverse NN* is to preprocess $P$ and create a data structure such that given a query point $q$, one can quickly return a set $S \subset P$ of size $k$ that lie within the radius $R$, while maximizing the diversity.

Formally, for any $q \in X$, let $B_P(q, R)$ be the set of points in $P$ within distance $R$ from $q$, and let OPT be a ($k', C$)-diverse set of $k^* = \min(k, |B_P(q, R)|)$ points from $B_P(q, R)$ that maximizes $C$. Then the data structure outputs ALG of size $k^*$ that is ($k', C/a$)-diverse such that $\mathsf{ALG_{k^*}} \leq cR$.

As described in the introduction, the problem addressed in the prior work (Abbar et al., 2013b) is the dual diverse NN problem, where the only consider $k' = 1$.

**Results.** Our main theoretical result is captured by the following theorems, which specifies the approximation and running time guarantees for our algorithms solving the primal and dual versions of the diverse nearest neighbor problem.

**Theorem A.5** (Primal Diverse ANN). *Let* $\mathsf{OPT} = \{p_1^*, ..., p_k^*\}$ *be a* ($k', C$)-*diverse solution that minimizes* $\mathsf{OPT_k}$. *Given the graph constructed by Algorithm 3, the search Algorithm 4 finds a* ($k', C/12$)-*diverse solution* ALG *with* $\mathsf{ALG_k} \leq \left(\frac{\alpha+1}{\alpha-1} + \epsilon\right) \mathsf{OPT_k}$ *in* $O\left(k \log_\alpha \frac{\Delta}{\epsilon}\right)$ *steps, where each step takes* $O\left((k^3/k')(8\alpha)^d \log \Delta\right)$ *time. The data structure uses space* $O(n(k/k')(8\alpha)^d \log \Delta)$.

**Theorem A.6** (Dual Diverse ANN). *Given the graph constructed by Algorithm 3, the search Algorithm 5 finds a* ($k', C/24$)-*diverse NN solution* ALG *satisfying* $ALG_k \leq \left(\frac{\alpha+1}{\alpha-1} + \epsilon\right) \cdot R$ *in* $\tilde{O}\left((k^4/k')(8\alpha)^d \log^2 \frac{\Delta}{\epsilon}\right)$ *time, if there exists a* ($k', C$)-*diverse solution* OPT *with* $\mathsf{OPT_k} \leq R$.

## A.2. Algorithm

**The preprocessing algorithm.** The indexing algorithm, which is the same for both the primal and dual versions of the problem, is shown in Algorithm 3. Line 12 of the algorithm uses the greedy algorithm of (Gonzalez, 1985), defined below.

---

**Algorithm 3** Indexing algorithm for diverse NN

---

1: **Input**: A set of $n$ points $P = \{p_1, ..., p_n\}$; $k$ is the size of the output; $k'$ is the parameter in the diversity definition; $\alpha$ is the parameter used for pruning.
2: **Output**: A directed graph $G = (V, E)$ where $V = \{1, ..., n\}$ are associated with the point set $P$.
3: **for** each point $p \in P$ **do**
4:     Sort all points $u \in P$ based on their distance from $p$ and put them in a list $\mathcal{L}$ in that order
5:     **while** $\mathcal{L}$ is not empty **do**
6:         $u \leftarrow \underset{u \in \mathcal{L}}{\operatorname{argmin}} D(u, p)$
7:         Initialize bag$[u] \leftarrow \{u\}$
8:         **for** each point $v \in \mathcal{L}$ in order **do**
9:             **if** $D(u, v) \leq D(p, u)/(2\alpha)$ **then**
10:                 bag$[u] \leftarrow$ bag$[u] \cup v$
11:                 remove $v$ from $\mathcal{L}$
12:             **end if**
13:         **end for**
14:         rep$[u] \leftarrow$ use the greedy algorithm of Gonzales to choose $k/k'$ points in bag$[u]$ to approximately maximize the minimum pairwise diversity.
15:         add edges from $p$ to rep$[u]$
16:         Remove $u$ from $\mathcal{L}$
17:     **end while**
18: **end for**

---

**Gonzales' greedy algorithm.** Given a set of $n$ points and a parameter $m$, the algorithm picks $m$ points as follows. The first point is chosen arbitrarily. Then, in each of the next $m-1$ steps, the algorithm picks the point whose minimum distance w.r.t. $\rho$ to the currently chosen points is maximized. It is known (Gonzalez, 1985) that this algorithm provides a 2-approximation for the problem of picking a subset of size $m$ which maximizes the minimum pairwise diversity distance between the picked points. Moreover, the picked set has an anti-cover property which we will discuss in Proposition A.10.

**Primal Search Algorithm.** Algorithm 4 shows the search algorithm for the primal version of diverse nearest neighbor. The algorithm is analyzed in Section A.3. The initialization step of line 3, can be done using the following algorithm.

**The initialization step.** Given a set $P$ of $n$ points equipped with metric distance $\rho$, and parameters $k'$ and $k$, and lower bound diversity $C$, the goal is to pick a subset $S \subseteq P$ of size $k$ which is $(k', C/4)$ diverse or otherwise output that no $(k', C)$-diverse subset $S$ exists. We use the following algorithm

- Initialize SOL $= \emptyset$

- While there exists a point $p \in P$ such that the ball $B = B_\rho(p, C/4)$ has $k'$ points in it, (i.e., $|B \cap P| > k'$),

    - Add an arbitrary subset of $B \cap P$ of size $k'$ to SOL.
    - Remove all points in $2B = B_\rho(p, C/2)$ from $P$.

- Add all remaining points in $P$ to SOL.

- If $|$SOL$| \geq k$, return an arbitrary subset of it of size $k$, otherwise return 'no solution'.

**Lemma A.7** (Initialization). *If $P$ has a subset* OPT *of size $k$ that is $(k', C)$-diverse, our initialization algorithm finds a $(k', C/4)$-diverse subset of size $k$.*

*Proof.* Note that it is straightforward to see why the set SOL that we get at the end is $(k', C/4)$-diverse. This is because first of all, each time we pick $k'$ points in a ball $B$ and add them to SOL, we make sure that no additional point will ever be

---

**Algorithm 4** Search algorithm for primal diverse NN

---

1: **Input**: A graph $G = (V, E)$ with $N_{out}(p)$ denoting the out edges of $p$; query $q$, number of optimization steps $T$; diversity lower bound $C$.
2: **Output**: A set of $k$ points ALG.
3: Initialize $\mathsf{ALG} = \{p_1, ..., p_k\}$ to be a set of $k$ points that are $(k', C/12)$-diverse using the initialization step proved in Lemma A.7.
4: **for** $i = 1$ to $T$ **do**
5:     $U \leftarrow \bigcup_{p \in \mathsf{ALG}} (N_{out}(p) \cup p)$, and sort $U$ based on their distance from $q$
6:     $\mathsf{ALG} \leftarrow$ the closest $k - 1$ points in ALG
7:     **for** each point $u \in U$ in order **do**
8:         **if** $\mathsf{ALG} \bigcup u$ is $(k', C/12)$-diverse **then**
9:             $\mathsf{ALG} \leftarrow \mathsf{ALG} \cup u$
10:        **end if**
11:        **if** $|\mathsf{ALG}| = k$ **then**
12:            Break
13:        **end if**
14:    **end for**
15: **end for**
16: **Return** ALG

---

picked in $2B$ and thus within distance $C/4$ of the points we pick there will be at most $k'$ points in the end. Second, at the end, every remaining ball of radius $C/4$ has less than or equal to $k'$ points in it. Therefore, we can pick all such points in the solution and everything we picked will be $(k', C/4)$ diverse.

Next we argue that we are in fact able to pick at least $k$ points in total which completes the argument. We do it by following the procedure of our algorithm and comparing it with OPT. At each iteration of the while loop that we remove $P \cap 2B$, we add exactly $k'$ points from $P \cap 2B$ to our solution SOL. Now note that the optimal solution OPT cannot have more than $k'$ points in $2B$ because by triangle inequality any pair of points in $2B$ have distance at most $C$, and picking more than $k'$ points in this ball contradicts the fact that OPT is $(k', C)$ diverse. Thus we can have an one-to-one mapping from each point in $\mathsf{OPT} \cap 2B$ to the $k'$ points in $P \cap 2B$ added to SOL. At the end of the while iteration, we know any unmapped point in OPT still exists in $P$, so we just map it to itself. By doing this, we can have an one-to-one mapping from OPT to SOL, which means that $|\mathsf{SOL}| \geq |\mathsf{OPT}| = k$. $\qquad\square$

**Dual Search Algorithm.** Algorithm 5 shows the search algorithm for the dual version of the diverse nearest neighbor problem. We provide the analysis in Section A.4.

### A.3. Analysis of the Primal Diverse NN Algorithm

In this section, we prove Theorem A.5 that gives the approximation and running time guarantees for Algorithm 3 and Algorithm 4.

**Lemma A.8.** *The graph constructed by Algorithm 3 has degree limit* $O((k/k')(8\alpha)^d \log \Delta)$.

*Proof.* Let's first bound the number of points not removed by others, then according to Line 14-15 in Algorithm 3, the degree bound will be that times $k/k'$.

We use $\mathsf{Ring}(p, r_1, r_2)$ to denote the points whose distance from $p$ is larger than $r_1$ but smaller than $r_2$. For each $i \in [\log_2 \Delta]$, we consider the $\mathsf{Ring}(p, D_{max}/2^i, D_{max}/2^{i-1})$ separately. According to Lemma 2.6, we can cover $\mathsf{Ring}(p, D_{max}/2^i, D_{max}/2^{i-1}) \cap P$ using at most $m \leq O((8\alpha)^d)$ small balls with radius $\frac{D_{max}}{2^{i+2}\alpha}$. According to the pruning criteria in Line 9, within each small ball, there will be at most one point remaining. This establishes the degree bound of $O((k/k')(8\alpha)^d \log \Delta)$. $\qquad\square$

**Lemma A.9.** *Suppose* $\mathsf{OPT} = \{p_1^*, ..., p_k^*\}$ *is a* $(k', C)$-*diverse solution with minimized* $\mathsf{OPT}_k$, *and let* $\mathsf{ALG} = \{p_1, ..., p_k\}$ *be the current solution (ordered by distance from* $q$). *If* $p_k \notin \mathsf{OPT}$, *there exists a point* $p^* \in \mathsf{OPT} \setminus \mathsf{ALG}$ *such that* $|B_\rho(p^*, C/2) \cap (\mathsf{ALG} \setminus p_k)| < k'$ *and* $\mathsf{ALG} \setminus p_k \bigcup p^*$ *is* $(k', C/4)$-*diverse.*

*Proof.* Recall that we use $B_\rho(p, r)$ to denote the ball in the $(X, \rho)$ metric space. Because $p_k \notin \mathsf{OPT}$, we have $\overline{\mathsf{OPT}} = \mathsf{OPT} \setminus \mathsf{ALG} \neq \varnothing$. We repeatedly perform the following operation until $\overline{\mathsf{OPT}}$ gets empty: select a point $p$ from $\overline{\mathsf{OPT}}$, and let $z = B_\rho(p, C) \cap \overline{\mathsf{OPT}}$, and remove $z$ from $\overline{\mathsf{OPT}}$. By doing this, we can get a list of points $\{p_1^*, ..., p_m^*\}$ and a partition of $\mathsf{OPT} \setminus \mathsf{ALG} = z_1 \cup z_2 ... \cup z_m$. By definition, we have the following properties:

- $\{p_1^*, ..., p_m^*\} \cap \mathsf{ALG} = \varnothing$

- $z_i \cap z_j = \varnothing$ for $i \neq j$

- $\sum_i |z_i| = |\mathsf{OPT} \setminus \mathsf{ALG}| = |\mathsf{ALG} \setminus \mathsf{OPT}|$

Now let $w_i = B_\rho(p_i^*, C/2) \cap (\mathsf{ALG} \setminus p_k \setminus \mathsf{OPT})$. Because all the $B_\rho(p_i^*, C/2)$ balls are disjoint, $\sum_i |w_i| \leq |\mathsf{ALG} \setminus p_k \setminus \mathsf{OPT}| < |\mathsf{OPT} \setminus \mathsf{ALG}| = \sum_i |z_i|$, there must exist an $i$ such that $|w_i| < |z_i|$. For that $i$, we have that $|B_\rho(p_i^*, C/2) \cap (\mathsf{ALG} \setminus p_k)|$ is equal to

$$\begin{aligned}
&= |B_\rho(p_i^*, C/2) \cap (\mathsf{ALG} \cap \mathsf{OPT})| + |B_\rho(p_i^*, C/2) \cap (\mathsf{ALG} \setminus p_k \setminus \mathsf{OPT})| && \text{(Because } p_k \notin \mathsf{OPT}) \\
&= |B_\rho(p_i^*, C/2) \cap (\mathsf{ALG} \cap \mathsf{OPT})| + |w_i| \\
&< |B_\rho(p_i^*, C/2) \cap (\mathsf{ALG} \cap \mathsf{OPT})| + |z_i| \\
&\leq |B_\rho(p_i^*, C/2) \cap (\mathsf{ALG} \cap \mathsf{OPT})| + |B_\rho(p_i^*, C) \cap (\mathsf{OPT} \setminus \mathsf{ALG})| \\
&\leq |B_\rho(p_i^*, C) \cap (\mathsf{ALG} \cap \mathsf{OPT})| + |B_\rho(p_i^*, C) \cap (\mathsf{OPT} \setminus \mathsf{ALG})| \\
&= |B_\rho(p_i^*, C) \cap \mathsf{OPT}| \\
&\leq k'
\end{aligned}$$

Therefore, we get $B_\rho(p_i^*, C/2) \cap (\mathsf{ALG} \setminus p_k) < k'$. Now, for any point $p \in B_\rho(p_i^*, C/4)$, $|B_\rho(p, C/4) \cap (\mathsf{ALG} \setminus p_k)| \leq |B_\rho(p_i^*, C/2) \cap (\mathsf{ALG} \setminus p_k)| < k'$, so we know that $\mathsf{ALG} \setminus p_k \cup p_i^*$ is $(k', C/4)$-diverse.

$\square$

The following is the well-known anti-cover property of the greedy algorithm of Gonzales whose proof we include for the sake of completeness.

**Proposition A.10.** *In Line 14 of Algorithm 3, let* $\mathsf{rep}[u]$ *be the output of greedily choosing $k/k'$ points in* $\mathsf{bag}[u]$ *maximizing minimum pairwise diversity. If a point $p \in \mathsf{bag}[u] \setminus \mathsf{rep}[u]$, we have* $\min_{v \in \mathsf{rep}[u]} \rho(p, v) \leq \min_{v_1, v_2 \in \mathsf{rep}[u]} \rho(v_1, v_2)$.

*Proof.* For the sake of contradiction, suppose $\min_{v \in \mathsf{rep}[u]} \rho(p, v) > \min_{v_1, v_2 \in \mathsf{rep}[u]} \rho(v_1, v_2)$, and the pairwise diversity minimizer is achieved by $\min_{v_1, v_2 \in \mathsf{rep}[u]} \rho(v_1, v_2) = \rho(x, y)$. Without loss of generality, we assume $x$ is added to $\mathsf{rep}[u]$ before $y$. At the time step $t$ when $y$ was added to $\mathsf{rep}_t[u]$, $\min_{v \in \mathsf{rep}_t[u]} \rho(y, v) = \rho(x, y)$ and $\min_{v \in \mathsf{rep}_t[u]} \rho(p, v) \geq \min_{v \in \mathsf{rep}[u]} \rho(p, v) > \rho(x, y)$, so $y$ wouldn't have been chosen by the greedy algorithm. Therefore, we have derived a contradiction. $\square$

**Lemma A.11.** *There always exists a point $p'$ connected from some point $w \in \mathsf{ALG}$ such that*

1. $\mathsf{ALG} \setminus p_k \bigcup p'$ *is $(k', C/12)$-diverse*

2. $D(p', q) \leq D(p_k, q)/\alpha + \mathsf{OPT}_k(1 + 1/\alpha)$

*Proof.* According to Lemma A.9, for any current solution $\mathsf{ALG}$ with $p_k \notin \mathsf{OPT}$, there exists a point $p^* \in \mathsf{OPT} \setminus \mathsf{ALG}$ such that $\mathsf{ALG} \setminus p_k \cup p^*$ is $(k', C/4)$-diverse. Let $w \in \mathsf{ALG}$ be the closest point to $p^*$. If there exists an edge from $w$ to $p^*$, replacing $p_k$ with $p^*$ is a potential update. We set $p' = p^*$ and $D(p', q) \leq \mathsf{OPT}_k$ satisfies the distance upper bound above.

Otherwise, we let $u$ be the point where $p^* \in \mathsf{bag}[u]$ but not selected into $\mathsf{rep}[u]$. For any point $p' \in \mathsf{bag}[u]$, $D(p', u) < D(w, u)/(2\alpha)$, so $D(p', p^*) < D(w, u)/\alpha < D(w, p^*)$. This means that all points in $\mathsf{bag}[u]$ are closer to $p^*$ than $w$, so they can't belong to $\mathsf{ALG}$. In the following, we consider two cases depending on whether $\min_{v \in \mathsf{rep}[u]} \rho(p^*, v) \geq C/3$. In each case, we will find a desired $p' \in \mathsf{rep}[u]$ and it is connected to $w$.

1. $\min_{v \in \text{rep}[u]} \rho(p^*, v) < C/3$: In this case, there exists another point $p' \in \text{rep}[u]$ with $D(p^*, p') \le D(p^*, u) + D(u, p') \le D(w, u)/\alpha$ and $\rho(p^*, p') < C/3$. Because $|B_\rho(p^*, C/2) \bigcap (\text{ALG} \setminus p_k)| < k'$, we have $|B_\rho(p', C/6) \bigcap (\text{ALG} \setminus p_k)| \subseteq |B_\rho(p^*, C/2) \bigcap (\text{ALG} \setminus p_k)| < k'$, so the addition of such $p'$ satisfies that $\text{ALG} \setminus p_k \cup p'$ is $(k', C/12)$-diverse.

2. $\min_{v \in \text{rep}[u]} \rho(p^*, v) \ge C/3$: In this case, according to Proposition A.10, we have $\text{rep}[u] = \{z_1, ..., z_{k/k'}\} \subseteq B(u, D(u, w)/(2\alpha))$ all with diversity distance at least $C/3$ from each other. Therefore, for any $p_i \in \text{ALG} \setminus p_k$, there can't exist two $z_j$ and $z_{j'}$ s.t. $\rho(p_i, z_j) < C/6$ and $\rho(p_i, z_{j'}) < C/6$. By a counting argument, we can find at least one $z_i$ s.t. $|B_\rho(z_i, C/6) \cap (\text{ALG} \setminus p_k)| < k'$. Finally, we let $p' = z_i$ where $\text{ALG} \setminus p_k \cup p'$ is $(k', C/12)$-diverse.

We have proved that the $p'$ we found satisfies the $(k', C/12)$-diverse criteria. Now we will bound its distance upper bound.

$$
\begin{aligned}
D(p', q) &\le D(p^*, q) + D(p', p^*) \le D(p^*, q) + D(p', u) + D(p^*, u) \\
&\le D(p^*, q) + D(w, u)/(2\alpha) + D(w, u)/(2\alpha) &\text{(Line 9 in Algorithm 3)} \\
&\le D(p^*, q) + D(w, u)/\alpha \\
&\le D(p^*, q) + D(w, p^*)/\alpha &\text{(Because } u \text{ is ordered earlier than } p^*) \\
&\le D(p^*, q) + D(w, q)/\alpha + D(p^*, q)/\alpha \le D(p_k, q)/\alpha + \text{OPT}_k(1 + 1/\alpha)
\end{aligned}
$$

$\square$

*Proof of Theorem A.5.* Regarding the running time, the total number of edges connected from any point in ALG is bounded by $|U| \le O((k^2/k')(8\alpha)^d \log \Delta)$. In each step, the algorithm first sorts all these edges and then checks whether each of them can be added to the new ALG set. The total time spent per step is $O(k|U| + |U| \log |U|)$. Usually, we assume $k \gg \log |U|$, and we can have the overall time complexity to be $O\left((k^3/k')(8\alpha)^d \log \Delta\right)$ per step.

To analyze the approximation ratio, at time step $t$, we use $\text{ALG}^\text{t} = \{p_1^t, ..., p_k^t\}$ to denote the current *unordered* solution. We denote $\text{ALG}_\text{k}^\text{t} = \max_{i \in [k]} D(p_i^t, q)$. According to Algorithm 4 and Lemma A.11, if $p_i$ is updated at time step $t$, we have $D(p_i^t, q) \le D(p_i^{t-1}, q)/\alpha + \text{OPT}_k(1 + 1/\alpha)$. By an induction argument, if a point $p_i$ is updated by $t$ times at the end of time step $T$, we have $D(p_i^T, q) \le \frac{D(p_i^0, q)}{\alpha^t} + \frac{\alpha+1}{\alpha-1}\text{OPT}_k$.

We now prove that $\text{ALG}_\text{k}^\text{T} \le \max_i \frac{D(p_i^0, q)}{\alpha^{T/k}} + \frac{\alpha+1}{\alpha-1}\text{OPT}_k$. Let $i \in [k]$ be the index achieving the maximal distance upper bound. For the sake of contradiction, if $\text{ALG}_\text{k}^\text{T} > \frac{D(p_i^0, q)}{\alpha^{T/k}} + \frac{\alpha+1}{\alpha-1}\text{OPT}_k$, this means that $p_i^T$ was updated for at most $T/k - 1$ times. By a counting argument, there exists another index $j$ which was updated for at least $T/k + 1$ times. However, at the time $t$ when $p_j^t$ was already updated for $T/k$ times, $D(p_j^t, q) \le \frac{D(p_j^0, q)}{\alpha^{T/k}} + \frac{\alpha+1}{\alpha-1}\text{OPT}_k < \text{ALG}_\text{k}^\text{T} \le \text{ALG}_\text{k}^\text{t}$, so the algorithm wouldn't have chosen $p_j^t$ to optimize cause it couldn't have had the maximal distance at that time, leading to a contradiction. Therefore, we prove that $\text{ALG}_\text{k}^\text{T} \le \max_i \frac{D(p_i^0, q)}{\alpha^{T/k}} + \frac{\alpha+1}{\alpha-1}\text{OPT}_k$.

Now we consider the following three cases depending on the value of the maximal $D(p_i^0, q)$. The case analysis here is similar to the proof in Theorem 3.4 from (Indyk & Xu, 2023).

**Case 1:** $D(p_i^0, q) > 2D_{max}$. Let $p_k^*$ be the point having the maximal distance from $q$ in an optimal solution OPT. We know that for any $p_i^0$, we have $D(p_k^*, q) \ge D(p_i^0, q) - D(p_i^0, p_k^*) \ge D(p_i^0, q) - D_{max} \ge D(p_i^0, q)/2$. Therefore, the approximation ratio after $T$ optimization steps is upper bounded by $\frac{\text{ALG}_\text{k}^\text{T}}{D(p_k^*, q)} \le \frac{D(p_i^0, q)}{D(p_k^*, q)\alpha^{T/k}} + \frac{\alpha+1}{\alpha-1} \le \frac{2}{\alpha^{T/k}} + \frac{\alpha+1}{\alpha-1}$. A simple calculation shows that we can get a $(\frac{\alpha+1}{\alpha-1} + \epsilon)$ approximate solution in $O(k \log_\alpha \frac{2}{\epsilon})$ steps.

**Case 2:** $D(p_i^0, q) \le 2D_{max}$ and $\text{OPT}_k > \frac{\alpha-1}{4(\alpha+1)}D_{min}$. To satisfy $\frac{D(p_i^0, q)}{\alpha^{T/k}} + \frac{\alpha+1}{\alpha-1}\text{OPT}_k \le (\frac{\alpha+1}{\alpha-1} + \epsilon)\text{OPT}_k$, we need $\frac{D(p_i^0, q)}{\alpha^{T/k}} \le \epsilon\text{OPT}_k$. Applying the lower bound $\text{OPT}_k \ge \frac{\alpha-1}{4(\alpha+1)}D_{min}$, we can get that $T \ge k \log_\alpha \frac{2(\alpha+1)\Delta}{(\alpha-1)\epsilon}$ suffices.

**Case 3:** $D(p_i^0, q) \le 2D_{max}$ and $\text{OPT}_k \le \frac{\alpha-1}{4(\alpha+1)}D_{min}$. In this case, we must have $k = 1$, because otherwise $D(p_k^*, p_1^*) \le 2D(p_k^*, q) < D_{min}$, violating the definition of $D_{min}$. Suppose $k = 1$ and the problem degenerates to the standard nearest neighbor search problem. After $T$ optimization steps, if $p_1^T$ is still not the exact nearest neighbor, we have

$D(p_1^T, q) \geq D(p_1^T, p_1^*) - \mathsf{OPT}_1 \geq \frac{D_{min}}{2}$. Applying the upper bound of $D(p_1^T, q)$ and $\mathsf{OPT}_1$, we have $\frac{D_{min}}{2} \leq D(p_1^T, q) \leq \frac{D(p_1^0, q)}{\alpha^T} + \frac{\alpha+1}{\alpha-1}\mathsf{OPT}_1 \leq \frac{D(p_1^0, q)}{\alpha^T} + \frac{D_{min}}{4}$. This can happen only if $T \leq \log_\alpha \frac{\Delta}{8}$. $\qquad \square$

### A.4. Analysis for the Dual Diverse NN Algorithm

In this section we analyze Algorithm 5.

---

**Algorithm 5** Search algorithm for dual diverse NN

---

1: **Input**: A graph $G = (V, E)$ with $N_{out}(p)$ denoting the out edges of $p$; query $q$; distance bound $R$; distance approximation error $\epsilon$.
2: **Output**: A set of $k$ points ALG.
3: Use binary search to find a maximal $C$ such that the initialization step proved in Lemma A.7 outputs a $(k', C)$-diverse set $\mathsf{ALG} = \{p_1, ..., p_k\}$
4: $\overline{C} \leftarrow 4C$
5: **while** $\max\limits_{p \in \mathsf{ALG}} D(p, q) > (\frac{\alpha+1}{\alpha-1} + \epsilon) \cdot R$ **do**
6:     $\overline{C} \leftarrow \overline{C}/2$
7:     **for** $i = 1$ to $c \cdot k \log_\alpha \frac{\Delta}{\epsilon}$ **do**
8:         $U \leftarrow \bigcup\limits_{p \in \mathsf{ALG}} (N_{out}(p) \cup p)$ and sort $U$ based on their distance from $q$
9:         $\mathsf{ALG} \leftarrow$ the closest $k - 1$ points in ALG
10:         **for** each point $u \in U$ in order **do**
11:             **if** $\mathsf{ALG} \bigcup u$ is $(k', \overline{C}/12)$-diverse **then**
12:                 $\mathsf{ALG} \leftarrow \mathsf{ALG} \cup u$
13:                 Break
14:             **end if**
15:         **end for**
16:     **end for**
17: **end while**
18: **Return** ALG

---

*Proof of Theorem A.6.* After applying the binary search to the initialization algorithm in Lemma A.7, we get an initial $(k', C)$-diverse solution and we know there doesn't exist a $(k', 4C)$-diverse solution. Therefore, we set $\overline{C} = 4C$ to be the upper bound on the maximal diversity we can achieve.

Then our Algorithm 5 is basically adding a binary search to Algorithm 4. Invoking the analysis from Theorem A.5, if there exists a $(k', C)$-diverse solution $\mathsf{OPT} = \{p_1^*, ..., p_k^*\}$ with $\mathsf{OPT}_k \leq R$, we can find a $(k', C/12)$-diverse solution $\mathsf{ALG} = \{p_1, ..., p_k\}$ with $ALG_k \leq \left(\frac{\alpha+1}{\alpha-1} + \epsilon\right) \cdot R$ in $O(k \log_\alpha \frac{\Delta}{\epsilon})$ steps where each step takes $\tilde{O}((k^3/k')(8\alpha)^d \log \Delta)$ time. As a result, each time when the algorithm enters the while loop on Line 5 in Algorithm 5, we know that there doesn't exist a $(k', \overline{C})$-diverse solution with maximal distance smaller than $R$. When we exit the while loop, the current $\overline{C}$ value is at least $1/2$ of the optimal $C$ value, and the current ALG solution we get is at least $(k', C/24)$-diverse. $\qquad \square$

## B. Algorithm Implementation

To conduct our experiments, we provide the heuristic algorithm that we designed for the $k'$-colorful nearest neighbor problem, based on the provable algorithms provided in the main paper. The provable indexing algorithm (3) has a runtime which is quadratic in the size of the data set and is slow in practice. This situation mimics the original DiskANN algorithm (Jayaram Subramanya et al., 2019), where the "slow preprocessing" algorithm has provable guarantees (Indyk & Xu, 2023) but quadratic running time, and was replaced by a heuristic "fast preprocessing" algorithm used in the actual implementation (Jayaram Subramanya et al., 2023). Here, Algorithm 9 offers a fast method tailored for the $k'$-colorful case, using several heuristics to improve the runtime. In the following section, we present the pseudocode for the procedures: search, index build, and the pruning procedure required for the index build.

**Diverse Search.** Our diverse search procedure, is a greedy graph-based local search method. In our search method, in each step, we maintain a list of best and diverse nodes, ensuring that at most $k'$ points are selected in the list per color. In each iteration of our search algorithm, we choose the best unexplored node and examine its out neighbors. From the union of our current list and the out neighbors, we select the best diverse set of nodes while satisfying the $k'$-colorful diversity constraint—meaning no color can have more than $k'$ points in the updated list. To identify the optimal diverse set from the union, we use a priority queue designed to accommodate the diversity constraint. Below, we present the pseudocode for this diverse priority queue.

---

**Algorithm 6** Insert $(p, d, c)$ into DiversePriorityQueue (Q, $L$, $k'$)

---

1: **Input**: Current queue $Q$, tuple $(p, d, c)$ of (point, distance, color) for new insertion, maximum size $L$ of the queue, maximum size $k'$ per color.
2: **Output**: Updated queue $Q$ after inserting $(p, d, c)$ which maintains the best set of at most $L$ points and at most $k'$ points of each color.
3: Let count$(c) \leftarrow$ number of elements in $Q$ with the color $c$.
4: Let maxDist$(c) \leftarrow$ maximum distance of an element in $Q$ with color $c$.
5: **if** count$(c) < k'$ **or** $d <$ maxDist$(c)$ **then**
6:     Insert $(p, d, c)$ into $Q$
7:     **if** count$(c) > k'$ **then**
8:         Remove the element with the maximum distance in $Q$ having color $c$.
9:     **end if**
10: **end if**
11: **if** $|Q| > L$ **then**
12:     Remove the element with the maximum distance in $Q$.
13: **end if**

---

Building on the previous explanation of the diverse priority queue, we outline the description of our diverse search procedure as follows.

---

**Algorithm 7** DiverseSearch$(G, s, q, k', k, L)$

---

1: **Input**: A directed graph $G$, start node $s$, query $q$, max per color parameter $k'$, search list size $L$.
2: **Output**: A set of $k$ points such that there are at most $k'$ points from any color.
3: Initialize DiversePriorityQueue $\mathcal{L} \leftarrow \{(s, D(s, q), col[s])\}$ with color parameter $k'$ and size parameter $L$.
4: Initialize a set of expanded nodes $\mathcal{V} \leftarrow \emptyset$
5: **while** $\mathcal{L} \setminus \mathcal{V} \neq \emptyset$ **do**
6:     Let $p^* \leftarrow \underset{p \in \mathcal{L} \setminus \mathcal{V}}{\operatorname{argmin}} D(p, q)$
7:     $\mathcal{V} \leftarrow \mathcal{V} \cup \{p^*\}$
8:     Insert $\{(p, D(p, q), col[p]) : p \in N_{\text{out}}(p^*)\}$ to $\mathcal{L}$
9: **end while**
10: **Return** [top $k$ NNs from $\mathcal{L}$; $\mathcal{V}$]

---

**Diverse Prune.** A key subroutine in our index-building algorithm is the prune procedure. Given a node $p$ and a set of potential outgoing edges $\mathcal{V}$, the standard prune procedure removes an edge to a vertex $w$ if there exists a vertex $u$ such that an edge $p \rightarrow u$ exists and the condition $D(u, w) \leq \frac{D(p,w)}{\alpha}$ is satisfied. Intuitively, this means that to reach $w$, we would first reach $u$, thus making multiplicative progress and eliminating the need for the edge $p \rightarrow w$, which contributes to the sparsity of the graph.

However, to account for diversity, the outgoing edges from the node must also be diverse and enable access to multiple colors. To address this requirement, we modify the standard prune procedure to incorporate the diversity constraint. The details of our revised algorithm are provided next.

**Diverse Index.** Our indexing algorithm follows the same approach as the DiskANN "fast preprocessing" heuristic implementation (Jayaram Subramanya et al., 2023), but we replace the search and prune procedures in their implementation

---

**Algorithm 8** DiversePrune$(p, \mathcal{V}, \alpha, R, m)$

---

1: **Input**: A point $p$, set $\mathcal{V}$, prune parameter $\alpha$, degree parameter $R$, and diversity parameter $m$.
2: **Output**: A subset $\mathcal{V}' \subseteq \mathcal{V}$ of cardinality at most $R$ to which edges are added.
3: Sort all points $u \in \mathcal{V}$ based on their distances from $p$ and add them to list $\mathcal{L}$ in that order.
4: Initialize sets blockers$[u] \leftarrow \emptyset$ for each $u \in \mathcal{V}$.
5: **while** $\mathcal{L}$ is not empty **do**
6:    $u \leftarrow \underset{u \in \mathcal{L}}{\operatorname{argmin}}\, D(u, p)$
7:    $\mathcal{V}' \leftarrow \mathcal{V}' \cup \{u\}$ and $\mathcal{L} \leftarrow \mathcal{L} \setminus \{u\}$
8:    **if** $|\mathcal{V}'| = R$ **then**
9:       **break**
10:   **end if**
11:   **for** each point $w \in \mathcal{L}$ **do**
12:      **if** $D(u, w) \leq D(p, w)/\alpha$ **then**
13:         blockers$[w] \leftarrow$ blockers$[w] \cup \{col(u)\}$
14:         **if** $|$blockers$[w]| = m$ or $col(u) = col(w)$ **then**
15:            $\mathcal{L} \leftarrow \mathcal{L} \setminus \{w\}$
16:         **end if**
17:      **end if**
18:   **end for**
19: **end while**
20: **Return** $\mathcal{V}'$

---

with our diverse search and diverse prune procedures. The details of our index-building procedure are provided below.

---

**Algorithm 9** DiverseIndex$(P, \alpha, L, R, m)$

---

1: **Input**: A set of $n$ points $P = \{p_1, \ldots, p_n\}$, prune parameter $\alpha$, search list size $L$, degree parameter $R$, and diversity parameter $m$.
2: **Output**: A directed graph $G$ over $P$ with out-degree at most $R$.
3: Let $s$ denote the estimated medoid of $P$.
4: Initialize $G$ with start node $s$.
5: **for** each $p_i \in P$ **do**
6:    Let $[\mathcal{L}; \mathcal{V}] \leftarrow$ DiverseSearch $(G, s, p_i, k' = L/m, L, L)$
7:    Let $\mathcal{V}' =$ DiversePrune $(p_i, \mathcal{V}, \alpha, R, m)$.
8:    Add node $p_i$ to $G$ and set $N_{\text{out}}(p_i) = \mathcal{V}'$ (out-going edges from $p_i$ to $\mathcal{V}'$).
9:    **for** $p \in N_{out}(p_i)$ **do**
10:      Update $N_{\text{out}}(p) \leftarrow N_{\text{out}}(p) \cup \{p_i\}$.
11:      **if** $|N_{\text{out}}(p)| > R$ **then**
12:        **Run** DiversePrune$(p, N_{\text{out}}(p), \alpha, R, m)$ to update out-neighbors of $p$.
13:      **end if**
14:    **end for**
15: **end for**

---

## C. Additional Plots

In this section we include the plots that were removed from the main body due to the space constraint.

### C.1. Remaining Plots from Section 4.2

Here we include the results plots for the two datasets SIFT-Skewed and SIFT-Balanced with $k' = 1$ and $k' = 10$ in Figures 5 and 6.

### C.2. Plots from Parameter Ablation Section 4.3

Here we include the plots for Recall versus Latency, for various diversity parameter $m$.

## D. DiskANN Overview

In this section we give an overview of the DiskANN procedures. For the full description the reader is referred to the original paper (Jayaram Subramanya et al., 2019).

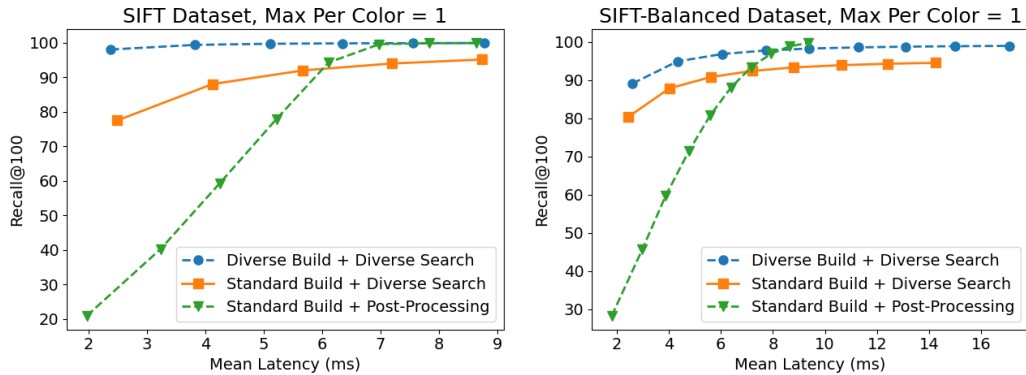

Figure 5: Recall vs Latency for SIFT-Skewed (left) and SIFT-Balanced (right) datasets with $k' = 1$.

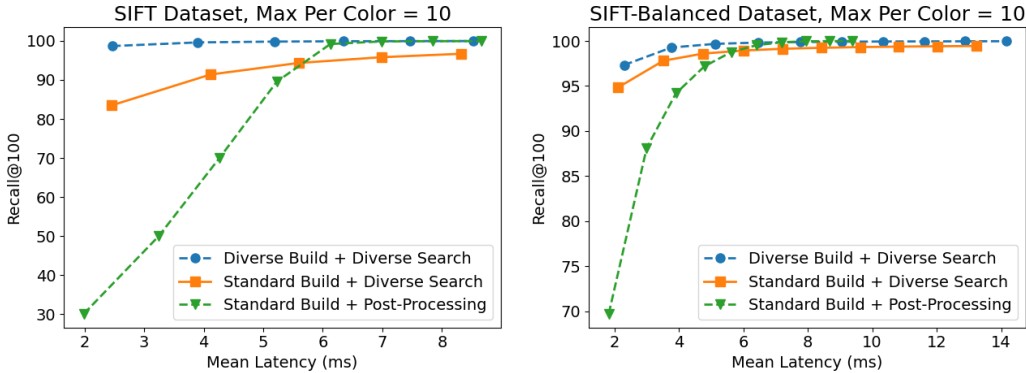

Figure 6: Recall vs Latency for SIFT-Skewed (left) and SIFT-Balanced (right) datasets with $k' = 10$.

The DiskANN data structure utilizes a directed graph $G$ with vertices $V$ corresponding to a set of points $P$. Once the graph is constructed, the search procedure begins from a specific vertex $s$ to answer a given query $q$. In the following, we provide a description of the search and insertion process.

The search procedure (Algorithm 11), $GreedySearch(s, q, L)$, takes the following parameters: the starting vertex $s$, the query point $q$, and the queue size $L$. It executes a best-first search using a queue of bounded length $L$, continuing until the $L$ vertices $v$ with the smallest values of $D(v, q)$ seen so far are all scanned. Upon completion, it returns a list of vertices ordered by increasing distance from $q$, where the first vertex (or the first $k$ vertices) are answers for the query. Note that the procedure will always run for at least $L$ steps, as long as the graph is connected. The total running time of the procedure is bounded by the number of steps multiplied by the out-degree bound of the graph $G$.

The construction of the graph $G = (V, E)$ is carried out through repeated calls to a procedure named $RobustPruning$ (Algorithm 10). Given a vertex $v$, a set of vertices $U$ (to be defined later), and parameters $\alpha > 1$ and $R$, the procedure $RobustPruning(v, U, \alpha, R)$ operates as follows. First, the set $U$ is sorted in ascending order of distance to $v$. The algorithm then iterates through this sorted list. Upon processing a new vertex $u$, it removes all other vertices $w$ from $U$ for which $D(u, w) \cdot \alpha < D(v, w)$. Finally, the vertex $v$ is connected to all remaining vertices in $U$ that have not been removed.

The starting point of the DiskANN data structure construction algorithm is the following simple procedure: for each vertex $v$, execute $RobustPruning(v, U, \alpha, R)$ with $U = V$ and $R = n$. In other words, robust pruning is performed on *every* vertex in the graph. We call this approach *slow preprocessing* (Algorithm 13), as a naive implementation incurs a time complexity of $O(n^3)$. Despite the high construction cost, (Indyk & Xu, 2023) demonstrate that this method reliably produces a graph whose degree grows only logarithmically with the graph's aspect ratio (under the assumption of constant doubling dimension), while ensuring that the greedy search procedure executes in polylogarithmic time.

Since the slow-preprocessing-algorithm is too slow in practice, the authors of (Jayaram Subramanya et al., 2019) propose a

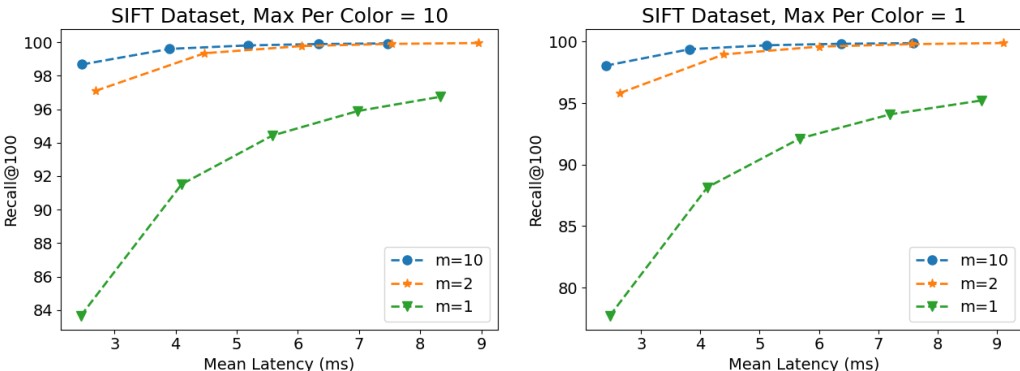

Figure 7: Recall vs Latency for SIFT-Skewed dataset with $k' = 10$ (left) and $k' = 1$ (right) by varying the diversity parameter $m$ during index construction. Higher $m$ implies more diversity.

faster heuristic method to construct the graph $G$, which we call *fast preprocessing* method (Algorithm 12). Initially, the graph $G$ is set to a random $R$-regular graph. Then, the construction of the graph $G = (V, E)$ proceeds incrementally. The algorithm makes two passes over the point set in a random order. For each vertex $v$ encountered, it computes a candidate set $U = GreedySearch(s, x_v, L)$ (starting from some vertex $s$), and applies the pruning procedure on $U$ rather than the entire vertex set $V$, i.e., it executes $RobustPruning(v, U, \alpha, R)$. Once pruning is complete, the algorithm inserts edges $(v, u)$ and $(u, v)$ for all $u \in U$ retained by the pruning step. If the degree of any vertex $u \in U$ exceeds the threshold $R$, its outgoing neighbors are also pruned via $RobustPruning(u, N_{out}(u), \alpha, R)$. This heuristic method is implemented and empirically evaluated in (Jayaram Subramanya et al., 2019).

---

**Algorithm 10** RobustPruning$(i, U, \alpha, R)$

---

1: **Input** Vertex $i$, candidate neighbor set $U$, pruning parameter $\alpha$, degree limit $R$(default $R$ is $n$ if not given)
2: **Result** Update $N_{out}(i)$, the set of out-neighbors of $i$
3: $U \leftarrow U \cup N_{out}(i)$
4: $N_{out}(i) \leftarrow \varnothing$
5: **while** $U \neq \varnothing$ and $|N_{out}(i)| < R$ **do**
6: $\quad v \leftarrow \operatorname{argmin}_{v \in U} D(x_v, x_i)$
7: $\quad N_{out}(i) \leftarrow N_{out}(i) \cup v$
8: $\quad U \leftarrow U \setminus v$
9: $\quad U \leftarrow \{v' \in U : D(x_v, x_{v'}) \cdot \alpha > D(x_i, x_{v'})\}$
10: **end while**

---

---

**Algorithm 11** GreedySearch$(s, q, L)$

---

1: **Input** Graph $G = (V, E)$, seed $s$, query point $q$, queue length limit $L$
2: **Output** visited vertex list $U$
3: $A \leftarrow \{s\}$
4: $U \leftarrow \varnothing$
5: **while** $A \setminus U \neq \varnothing$ **do**
6:     $v \leftarrow \text{argmin}_{v \in A \setminus U} D(x_v, q)$
7:     $A \leftarrow A \cup N_{out}(v)$
8:     $U \leftarrow U \cup v$
9:     **if** $|A| > L$ **then**
10:        $A \leftarrow$ top $L$ closest vertices to $q$ in $A$
11:     **end if**
12: **end while**
13: sort $U$ in increasing distance from $q$
14: **return** $U$

---

---

**Algorithm 12** DiskANN indexing algorithm (with fast preprocessing)

---

1: **Input** Point set $P = \{x_1...x_n\}$, degree limit $R$, queue length $L$
2: **Output** A proximity graph $G = (V, E)$ where $V = \{1..n\}$ are associated with point sets $P$.
3: $G \leftarrow$ randomly sample a $R$-regular graph on vertex set $V = \{1..n\}$
4: $s \leftarrow$ vertex for the point closest to the centroid of $P$
5: **for** $k = 1$ **to** $2$ **do**
6:     $\sigma \leftarrow$ a random permutation of $[1...n]$
7:     **for** $i = 1$ **to** $n$ **do**
8:       $U \leftarrow GreedySearch(s, x_{\sigma(i)}, L)$
9:       $RobustPruning(\sigma(i), U, \alpha, R)$
10:       **for** vertex $j$ in $N_{out}(\sigma(i))$ **do**
11:         $N_{out}(j) \leftarrow N_{out}(j) \cup \sigma(i)$
12:         **if** $|N_{out}(j)| > R$ **then**
13:           $RobustPruning(j, N_{out}(j), \alpha, R)$
14:         **end if**
15:       **end for**
16:     **end for**
17: **end for**

---

---

**Algorithm 13** DiskANN indexing algorithm (with slow preprocessing)

---

1: **Input** Vertex set $P = \{x_1...x_n\}$, parameters: degree limit $R$
2: **Output** A proximity graph $G = (V, E)$ where $V = \{1..n\}$ are associated with point sets $P$.
3: $s \leftarrow$ vertex for the point closest to the centroid of $P$
4: **for** $i = 1$ **to** $n$ **do**
5:     $N_{out}(i) \leftarrow RobustPruning(i, V, \alpha, R)$
6: **end for**

---

