# OpenReview forum: "Graph-Based Algorithms for Diverse Similarity Search"
_ICML.cc/2025/Conference — ICML 2025 poster_

### Official Review · Reviewer_NUeQ · 2025-03-06

**Overall Recommendation:** 4

**Summary:**

This paper studies the problem of diversifying the results of approximate nearest search (ANNS) for vectors. It adapts the state-of-the-art proximity graph algorithm for ANNS and modifies both the graph construction and query processing algorithm to consider diversity. It also conducts theoretical analysis to show the query processing complexity of the proposed algorithms and extends the algorithms to general definitions of diversity. The empirical results show that the proposed algorithms significantly improves the performance compared with the original proximity graph algorithm.

**Claims And Evidence:**

Yes

**Essential References Not Discussed:**

No

**Experimental Designs Or Analyses:**

Yes. The experiment is sound.

**Methods And Evaluation Criteria:**

Yes

**Other Comments Or Suggestions:**

Following the weakness, I have two suggestions that may enhance the paper.

(1) In this paper, the goal is to minimize S_k, i.e., the kth nearest neighbor. [1] requires to maximizes the total similarity score of the k results (or equivalently, minimize the total distances). What are the relations of the two results requirements?
[1] Diversifying TopK Results

(2) One interesting instance of the diversified search problem is that the diversity measure is also the distance itself. For example, when handling image embeddings without labels, we may want the returned results to differ by a distance threshold to impose semantic distinctiveness. Could you add an experiment for this instance with the proposed algorithms?

**Other Strengths And Weaknesses:**

Thank for the interesting paper!

Strength

(1) Diversified ANNS is an important problem of practical significance.

(2) The proposed algorithms make sense, i.e., by considering diversity when adding the edges and staring search from a diverse set.

(3) Extensive theoretical analysis is conducted.

(4) More general cases of diversified search are disused and extended.

Weakness

(1) The relation w.r.t. alternative result quality definitions is not discussed.

(2) Some experiment results for general cases should be provided.

**Questions For Authors:**

NA

**Relation To Broader Scientific Literature:**

This paper studies diversified ANNS and may benefit applications that require result diversification.

**Theoretical Claims:**

Yes. The theorems are correct to my understanding.

---

> ### Author Rebuttal · Authors · 2025-03-27
>
> Thank you for your valuable feedback!
>
> **Response for W1/Q1:** Thanks for your suggestion! For the colorful NN definition, the current algorithm 2 always optimizes the furthest point, which only provides approximation guarantees on the maximal distance but not the total distance. For the total (average) distance, we have developed a modified search algorithm (not included in the paper)  that optimizes the k points together in a single step. For that algorithm we obtain a “uniform” guarantee $ALG_i\le (\frac{\alpha+1}{\alpha-1}+\epsilon)OPT_i$ for all $ i<k$ where $ALG_i$ and $OPT_i$ are the distance from the i-th point in our solution ALG and the optimal solution OPT. Note that the point-wise guarantee implies a bound on the total distance.
>
> **Response for W2/Q2:** Thanks for proposing this idea! This setting fits in our (k’,C)-diverse NN formulation in Definition A.3, so our algorithm can handle this setting, at least from a theoretical perspective. However, our codebase is optimized for the colorful NN problem (see Appendix B) because of the concrete application we targeted. Performing experiments under the general diversity setting requires a separately optimized algorithm implementation tailored to that case, which is possible but needs significant work.

---

### Official Review · Reviewer_iY1E · 2025-03-10

**Overall Recommendation:** 4

**Summary:**

This paper considers the nearest-neighbor search problem with diversity constraints. It builds on existing graph-based algorithms for similarity search and proposes a new indexing algorithm which can more efficiently answer queries with diversity constraints.

Through experimentation on several large datasets, the effectiveness of the new algorithm is demonstrated when compared with existing methods for diverse similarity search.

## Update after Rebuttal

I have nothing further to add following the rebuttal by the authors.

**Claims And Evidence:**

The claimed theoretical and empirical results are well backed up in the paper.

**Essential References Not Discussed:**

No.

**Experimental Designs Or Analyses:**

N/A

**Methods And Evaluation Criteria:**

The experimental evaluation methodology seems reasonable. It may be interesting to also compare with other nearest-neighbor algorithms beyond DiskANN.

**Other Comments Or Suggestions:**

None.

**Other Strengths And Weaknesses:**

The paper addresses a natural problem, and proposes an effective algorithm.

In my view, the experimental section could be more extensive by comparing against additional baseline methods for ANN and post-processing, such as HNSW.

Finally, one possible downside of this approach is the need to build the index for a specific diversity requirement: that is, for returning $k$ neighbors with at most $k'$ of the same color. The proposed algorithm could be even more general if a flexible index could support queries with varying diversity constraints.

**Questions For Authors:**

1. The index is built based on the values of $k$ and $k'$, and so they need to be known in advance. On the other hand, in the experimental section, you compare with using the 'diverse query' on the standard index. Is there some kind of intuitive interpolation between these cases, such a 'balanced index' which is effective at responding to non-diverse queries, and diverse queries whose parameters are not known in advance?

**Relation To Broader Scientific Literature:**

Approximate nearest neighbour search is a hugely important and active area of research, into which this work fits naturally. The diverse ANNS problem is well motivated in this paper, which puts it into the context of the literature.

**Theoretical Claims:**

The paper proves a bound on the running time complexity of the query algorithm, and gives an approximation guarantee for the returned near-neighbors. The theoretical claims appear sound.

---

> ### Author Rebuttal · Authors · 2025-03-27
>
> Thank you for your valuable feedback!
>
> > W1: In my view, the experimental section could be more extensive by comparing against additional baseline methods for ANN and post-processing, such as HNSW.
>
> **Response:** It is indeed possible that using different graph-based methods could improve the performance. However, this could hold for both for the baseline re-ranking and for the diversity-aware algorithm - we believe that other graph-based algorithms like HNSW should be amenable to the modifications we applied to DiskANN. Our strategy was instead to start from the existing DiskANN code and make as minimal changes to it as possible, in order to retain optimizations that have been already developed for that code. This allows us to make “apples to apples” comparisons, as we use the same code base for both re-ranking and our algorithm.
>
> > W2: Finally, one possible downside of this approach is the need to build the index for a specific diversity requirement: that is, for returning k neighbors with at most k′ of the same color. The proposed algorithm could be even more general if a flexible index could support queries with varying diversity constraints.
>
> **Response:** We note that, to obtain theoretical guarantees, it suffices to know an “upper bound” on the value of k. (It is easy to see that building a graph on a larger value of k works for searching on smaller values of k. We are happy to include this remark in the paper.)
> On the practical side, we use the tunable parameter ‘m’ which determines the number of colorful neighbors a candidate needs to have (defined in Section 4, line 325). Empirically,  we find that m can be set much smaller than k while retaining good results. In fact, in our experiments, we studied the performance of the algorithm even with m=1, i.e., where the graph construction method is the same as in the vanilla DiskANN but the search retrieves k>1 diverse candidates. Please see Figure 7 for this ablation study
>
> We also want to emphasize that in the worst case, the index must account for the number of retrieved elements (k) in the colorful nearest neighbor (NN) problem. Consider an extreme scenario where we color a dataset using $k^*$ colors, $k^*-1$ of which are uniformly and evenly assigned across the dataset, and the last color is randomly assigned to only one point. For the colorful NN query with $k \leq k^*-1$, we can ignore the last color, and a diverse search applied to the standard index should work. However, for $k = k^*$, the data structure needs to find a set of $k−1$ close points, plus the unique point with the $k$-th color. Without appropriate preprocessing, identifying a point with a unique color in the data set can take linear time. Therefore, the index-building algorithm should account for this and add more edges toward the point with the unique color. This justifies the need to know the value of $k$ during the preprocessing.
>
> > Q1: The index is built based on the values of k and  k′ and so they need to be known in advance. On the other hand, in the experimental section, you compare with using the 'diverse query' on the standard index. Is there some kind of intuitive interpolation between these cases, such a 'balanced index' which is effective at responding to non-diverse queries, and diverse queries whose parameters are not known in advance?
>
> **Response:** We believe that the ideal index should take into account how restrictive the diversity constraint is e.g. for the colorful case, the larger k is, the more restrictive the constraint is, so our index should keep more edges between different colors. Please see the intuitive analysis above. Our experiment shows that the diverse search algorithm also helps find more diverse answers even when applied to the standard index. In conclusion, both the diverse index and the diverse search help solve this problem.

---

### Official Review · Reviewer_qrKi · 2025-03-12

**Overall Recommendation:** 3

**Summary:**

In this paper, the authors present provably efficient algorithms for approximate nearest neighbor search with diversity constraints. They propose several problem formulations, such as Colorful NN and k'-Colorful NN, and further generalize them into more general problems. To solve these problems, the authors propose related indexing and search algorithms, providing a theoretical analysis of query results under the assumption of bounded doubling dimension. In experiments, they demonstrate that the proposed algorithm outperforms straightforward vector search followed by attribute filtering.

---

## Update After Rebuttal

The authors' rebuttal and the follow-up discussion addressed my questions. I will raise my score from 2 to 3.

**Claims And Evidence:**

Yes.

**Essential References Not Discussed:**

W1. The literature survey is insufficient. In the related work section, the authors mention only a few LSH papers while omitting similarity-graph (a.k.a. proximity-graph) papers. Given that this paper focuses on graph-based search, a more detailed introduction to graph-based methods is important.

**Experimental Designs Or Analyses:**

Yes, I believe the experimental results are reasonable.

**Methods And Evaluation Criteria:**

Yes, but some related methods are missing, see my comments below.

**Other Comments Or Suggestions:**

I recommend that the authors add sufficient literature on similarity graphs. See W1.

**Other Strengths And Weaknesses:**

### **Strength**

S1. This paper researched a new task in the category of ANNS with attribute constraints.

S2. The theoretical analysis of proposed algorithms is thorough and sound.

### **Weakness**

W2. I wonder whether the problem setting is sufficiently useful. Given that there are almost no baseline methods specifically designed for this task, could the authors provide more evidence on the usefulness of the central problem, namely, colorful NN?

W3. The baseline algorithms are insufficient. The reasons are summarized as follows.

- W3.1. The authors claimed that the first stage of retrieving a large number of points closest to the query is time-consuming, and I agree with this. However, some research has shown significant improvements in the efficiency of original similarity graphs [1][2]. Since these methods are orthogonal to vanilla DiskANN, combining them with the original DiskANN build and diverse search might yield better performance than the diverse build + diverse search approach.

- W3.2 The color is essentially an attribute, so this work falls into the category of ANNS with constrained attributes. Several research papers have addressed this problem, with a representative one being [3]. Could the authors compare diverse search with the approach in [3]? Although the attribute constraints are different, I wonder if the graph construction in [3] could be easily modified to solve the colourful NN tasks.

- W3.3 In the current design of baseline algorithms in the experiments, only the strategy of vector search followed by attribute filtering is considered. It is also recommended to consider the alternative strategy of attribute filtering followed by vector search. In fact, we can divide the original set into multiple subsets based on their colors, then use vector quantization to index points in small subsets and use similarity graphs to index points in large subsets. During the query phase, we can combine the search results from different subsets to obtain the final results. I wonder if the proposed method in this paper can outperform such a straightforward strategy.

[1] Patrick H. Chen, Wei-Cheng Chang, Jyun-Yu Jiang, Hsiang-Fu Yu, Inderjit S. Dhillon, Cho-Jui Hsieh: FINGER: Fast Inference for Graph-based Approximate Nearest Neighbor Search. WWW 2023.

[2] Kejing Lu, Chuan Xiao, Yoshiharu Ishikawa: Probabilistic Routing for Graph-Based Approximate Nearest Neighbor Search. ICML 2024.

[3] Mengzhao Wang, Lingwei Lv, Xiaoliang Xu, Yuxiang Wang, Qiang Yue, Jiongkang Ni. An Efficient and Robust Framework for Approximate Nearest Neighbor Search with Attribute Constraint. NeurIPS 2023.

**Questions For Authors:**

Please answer the questions or provide your comments to address the concerns raised in W1-W3.

**Relation To Broader Scientific Literature:**

The contribution may be limited, as the problem studied in the paper is essentially ANNS with attribute constraints, which has been explored in previous research.

**Theoretical Claims:**

Yes.

---

> ### Author Rebuttal · Authors · 2025-03-27
>
> Thank you for your valuable feedback!
>
> **Response to W1:** Since we cite HNSW, NGT and DiskANN earlier in the introduction, we did not want to repeat the discussion later.. However, given the feedback, we will expand the discussion and provide a comprehensive overview of graph-based NN algorithms.
>
> **Response to W2:** Diversity-based retrieval has found many applications since the influential paper of (Carbonell & Goldstein, 1998). To the best of our knowledge, the predominant algorithmic approach to this problem relied on re-ranking/filtering as proposed in that paper.
>
> As mentioned in our paper, Google emphasized the importance of “diversity” w.r.t intent during search to capture multiple intents. E.g., searching for “transformers” without other context should ideally return results related to the movie, the electrical engineering concept and the ML concept. We can capture this in our framework by using some classification algorithm to assign a color to each vector based on its underlying “intent”.
>
> Similarly, there is a recent body of literature which highlights the importance of diverse retrievals in Retrieval Augmented Generation (RAG) systems. For example, the paper https://arxiv.org/pdf/2409.18110 argues that retrievals need to be diverse w.r.t perspective to excel in RAG queries. Similarly, the work https://arxiv.org/pdf/2502.11228v1 also argues for diverse retrieval using Vendi scores. Conceptually speaking, both these works seek to ensure diversity during retrieval, which is exactly what our work is targeting, albeit with slightly different modeling. We therefore believe that our work, which studies the problem of efficient diverse retrievals, will prove to be important in furthering this line of inquiry.
>
> Regarding the specific formulation of colorful nearest neighbor, here are two specific real-world applications:
> 1. Seller diversity, which we include in the paper (please refer to e.g. line 295-right for the description).
> 2. Typically in RAG systems, documents are divided into chunks/passages and each passage is embedded as a vector. Therefore each vector in the data set is associated with a document (and has a docID which we see as a color). It has been noted in practice that when we retrieve top k vectors to the query, many vectors retrieved correspond to passages from the same document, making them redundant as the eventual goal is to pass the complete documents corresponding to the docIDs retrieved through the vector search. In such scenarios, retrieving vectors corresponding to diverse set of docIDs, would help improve the efficiency and accuracy of the RAG pipelines.
>
> **Response to W3.1:** It is indeed possible that integrating faster methods for selecting promising nodes to expand [1,2] into DiskANN could improve its performance. Our strategy was instead to start from the existing DiskANN code and make as minimal changes to it as possible, in order to retain optimizations that have been already developed for that code. In addition to retaining the efficiency benefits of those optimizations, this also allows us to make “apples to apples” comparisons, as we use the same code base for both re-ranking and our algorithm. It is plausible, although not guaranteed, that further optimizations along the lines of [1] and [2] would equally benefit both our algorithm and the re-ranking baseline.
>
> **Response to W3.2:** Thank you for mentioning reference [3], and it is an important distinction to make  - it appears that their framework is indeed substantially different from ours. Most notably, attribute constraints in [3] are “local”: they specify a relation (or filter constraint) between the query q and a data point p which must be satisfied if p is to be included in the output. In contrast, diversity constraints are “global”, as they specify the overall constraint on the whole output  (e.g., at most k’ out of k points can have the same color). This difference is reflected in the indexing procedures. In particular, the approach of [3] augments the distance function so that close points with similar attributes are linked in the graph. In contrast, the main goal of our indexing procedure is to connect each node to close points with dissimilar attributes (colors). We will include this discussion in the paper.
>
> **Response to W3.3:** The number of categories can be very large. In our seller data set, this number is roughly 2500, and in the Amazon Automotive dataset, this number is roughly 85000. Therefore, in practice, it can be quite inefficient to enumerate the prepared data structure for each color.

---

> > ### Comment · Reviewer_qrKi · 2025-04-02
> >
> > Thank you for your responses.
> >
> > The responses to W2 and W3.3 look good.
> >
> > As for W3.1 and W3.2, due to the absence of an experimental comparison, I am not sure if the proposed algorithm can perform better than the candidates [1][2][3] listed here. I suggest that the authors include a comparison in discussion to help readers understand in which scenario the proposed algorithm is most suitable.

---

> > > ### Author Response · Authors · 2025-04-06
> > >
> > > Thank you for the suggestion. In the final version of the paper, we will discuss the methods in these references, and how/whether they apply to the scenario in our paper. Let us clarify the rationale behind our current experimental choices and emphasize the broader scope of our contributions.
> > >
> > > - **Generality of our approach:** The key idea in our work—modifying graph construction to encourage diversity—is conceptually simple and broadly applicable. It can technically be integrated into any graph-based ANN algorithm that uses greedy search, such as HNSW, FINGER, or others. Our method does not rely on DiskANN-specific internals, and we expect our techniques will yield gains on top of these algorithms as well.
> > >
> > >
> > >
> > > - **Choice of DiskANN:** We chose to apply our ideas within the DiskANN framework because it is, to our knowledge, the only real-world graph-based ANN algorithm that comes with worst-case theoretical guarantees. This allows us to ground our contributions in a rigorous yet practically relevant setting.
> > >
> > >
> > >
> > > - **On comparisons with [1] and [2]:** These techniques focus on optimizing the graph traversal during greedy search by either ruling out expensive distance calculations of candidates which will not improve the priority queue (by cleverly using estimates in [1], and probabilistically excluding bad candidates using LSH-like methods in [2]), and are largely orthogonal to our contribution. Since both methods rely on greedy-like search, we believe that our diverse-search and diverse-build ideas will be useful, and have similar relative impact to the gains we observe for DiskANN.
> > >
> > >
> > >
> > > - **On comparison with [3]:** The problem addressed in [3]—ANN with local attribute constraints—is fundamentally different from our goal of ensuring global diversity in the returned set. In the [3], the goal is to obtain results with **similar metadata** as the query metadata (say color=RED or brand=NIKE as specified by the user query in a shopping scenario), whereas in our setting, there is no user specified constraint and the goal is to output relevant vectors with **sufficiently dissimilar metadata**. This crucial difference extends to both problem formulation as well as indexing and search strategies, and so we believe that direct experimental comparison may not yield meaningful insights.
> > >
> > > We will include a thorough discussion by expanding on the above points in the final version of the paper. We hope this helps readers better understand the unique value of our approach and when it may be preferred over alternatives.

---

### Official Review · Reviewer_JqK5 · 2025-03-12

**Overall Recommendation:** 3

**Summary:**

This paper addresses an important problem of graph-based nearest neighbor search (NNS) with diversity constraints. The new algorithm is proposed that is supported by theoretical analysis and also shows promising experimental results.

## update after rebuttal

I appreciate the authors' rebuttal and will keep my original score.

**Claims And Evidence:**

Claims made in the submission are supported by convincing evidence.

**Essential References Not Discussed:**

There are other papers on graph-based NNS with theoretical analysis that can be mentioned for completeness, e.g., (Laarhoven, 2018; Fu et al., 2019; Prokhorenkova & Shekhovtsov, 2020; Lu et al., 2024; Diwan et al., 2024).

**Experimental Designs Or Analyses:**

I haven’t found any major problems with experimental design.

**Methods And Evaluation Criteria:**

The proposed methods and evaluation are reasonable.

**Other Comments Or Suggestions:**

L83 (left): I suggest using “best-first search” instead of “greedy search” since usually several best candidates are maintained to reduce the probability of getting stuck in a local optimum.

$\alpha$ is not defined before first mentioned in line 66.

Some typos:
- There are multiple uses of \citep instead of \citet.
- L128 (left): footnotes are typically placed after punctuation marks.
- L121 (right): “note that for” -> “note that”
- L203 (left): “let’s” -> “let us”
- L414 (left): “Figures 5, and 6” ->  “Figures 5 and 6”
- L459 (left): typo in the url.

**Other Strengths And Weaknesses:**

The paper addresses an important problem in graph-based NNS. Importantly, it theoretically analyzes graphs constructed with neighbor pruning, which is known to be critical for effective graph-based NNS.


The weaknesses of the paper are the following:
1. Some presentation issues. For example, the paper is significantly based on the DiskANN algorithm, but the algorithm is not described which makes it more difficult to follow the paper. Informal descriptions of Algorithms 1 and 2 (in addition to the pseudocode) would also be helpful. There is no conclusion section in the paper.
2. Algorithm limitation: one disadvantage of the proposed algorithm is that one needs to know the number of retrieved elements ($k$) before constructing the graph.
3. Also, in some cases the simplest baseline outperforms other approaches (see, e.g., Figures 5 and 6).

**Questions For Authors:**

Q1. Line 195 (right): what is OPT_k?


Q2. Can there be other simple baselines, e.g., having separate indices for some categories?


Q3. In Standard DiskANN Build + Post-Processing, how is $r$ chosen? As I understand, both $r$ and $L$ affect the time-accuracy trade-off.


Q4. In some cases, the simplest baseline outperforms other approaches (see, e.g., Figures 5 and 6). What can be the reason for that? Can some heuristics be used to improve the performance of the proposed approach in such scenarios?
Q5. From Figure 7 it seems that higher values of diversity parameters are better.  Does this hold for larger values too?

**Relation To Broader Scientific Literature:**

The paper addresses an important problem in graph-based NNS that, to the best of my knowledge, has not been previously addressed.

**Theoretical Claims:**

I’ve only partially checked the proofs and haven’t found any issues.

---

> ### Author Rebuttal · Authors · 2025-03-27
>
> Thank you for your valuable feedback!
>
> Response for W1: Thank you for the suggestions.  We will include preliminaries on the DiskANN algorithm in the appendix. We have included intuition on the new algorithms, but for completeness we will add informal descriptions of them too. We will also include a conclusion to the paper
>
> Response for W2: We note that, to obtain theoretical guarantees, it suffices to know an “upper bound” on the value of k. (It is easy to see that building a graph on a larger value of k works for searching on smaller values of k. We are happy to include this remark in the paper.)
> On the practical side, we use the tunable parameter ‘m’ which determines the number of colorful neighbors a candidate needs to have (defined in Section 4, line 325). Empirically,  we find that m can be set much smaller than k while retaining good results. In fact, in our experiments, we studied the performance of the algorithm even with m=1, i.e., where the graph construction method is the same as in the vanilla DiskANN but the search retrieves k>1 diverse candidates. Please see Figure 7 for this ablation study
> We also want to emphasize that in the worst case, the index must account for the number of retrieved elements (k) in the colorful nearest neighbor (NN) problem. Consider an extreme scenario where we color a dataset using $k^*$ colors, $k^*-1$ of which are uniformly and evenly assigned across the dataset, and the last color is randomly assigned to only one point. For the colorful NN query with $k \leq k^*-1$, we can ignore the last color, and a diverse search applied to the standard index should work. However, for $k = k^*$, the data structure needs to find a set of $k−1$ close points, plus the unique point with the $k$-th color. Without appropriate preprocessing, identifying a point with a unique color in the data set can take linear time. Therefore, the index-building algorithm should account for this and add more edges toward the point with the unique color. This justifies the need of knowing the value of $k$ during the preprocessing.
>
> Response for W3: Please note that diverse-build+diverse-search is never outperformed by the baseline except in a tiny fraction of cases, and that too only by a small amount. Moreover, this occurs only in the very high latency regimes. A possible explanation for when and why this occurs could be the following: in datasets where there is more balance in the colors, the naive search itself organically retrieves diverse items which makes it easier for the baseline algorithm without the overheads of our slightly expensive diverse priority queue data structure. In a majority of cases (including on the real-world dataset), our approach is significantly superior to the baseline. Finally, real-world use-cases often deal with the lower latency regimes as long as the desired recall is achieved.
>
> Response for Q1: Please see Definition 2.3 for $OPT_k$. We will add a reference to the definition in line 195
>
> Response for Q2: The number of categories can be very large. In our seller data set, this number is roughly 2500, and in the Amazon Automotive dataset, this number is roughly 85000. Therefore, in practice, it can be quite inefficient to enumerate the prepared data structure for each color.
>
> Response for Q3: In the Standard DiskANN algorithm, there is one search time parameter, L, which is the size of the priority queue the algorithm maintains till the search converges. Indeed, upon convergence, the algorithm knows the local optimal set of L potential candidates. We merely set ‘r = L’, and select the most diverse subset of these L candidates as a post-processing step. Therefore, in this baseline, we can think of r = L and sweep over the values of L as is usually done. There is an overloaded priority queue size parameter L during index construction, which we fix to be 200, which is within the range of commonly chosen values (typically around 2 to 3 times the graph degree). We apologize for this notational mistake and can disambiguate to Ls and Lb to denote search-time vs build-time parameters.
>
> Response for Q4: We have suggested possible explanations for the reason why the baseline is marginally outperforming our approach. As for heuristics to improve, if there are scenarios where the baseline is significantly superior, one could then potentially have a pre-determined cut-off for the list-size ‘L’ parameter at search time, such that we implement our diverse search + diverse build algorithm for values of L below the cut-off, and we perform a post-processing approach on the diverse index built for values of L above the cut-off, which we have empirically validated to be superior to the baseline algorithm reported in the paper. As for the last question of whether Figure 7 holds for larger values too, we have tried much larger values as well, and it indeed holds that increasing ‘m’ yields better performance, but with diminishing returns as observed in the figure with values of m=2 and m=10.

---

### Decision · Program_Chairs · 2025-05-01

**Decision:**

Accept (poster)

**Comment:**

This paper proposes an efficient graph-based approach for nearest neighbor search. The reviewers appreciated that the paper addresses an important problem and that the proposed method is theoretically sound. Given the positive evaluations from the reviewers, I recommend acceptance.